# Elign: Equivariant Diffusion Model Alignment from Foundational Machine Learning Force Fields

**Yunyang Li** [1] **Lin Huang** [2] **Luojia Xia** [3] **Wenhe Zhang** [1] **Mark Gerstein** [1 3]

## Abstract

Generative models for 3D molecular conformations must respect Euclidean symmetries and concentrate probability mass on thermodynamically favorable, mechanically stable structures. However, E(3)-equivariant diffusion models often reproduce biases from semi-empirical training data rather than capturing the equilibrium distribution of a high-fidelity Hamiltonian. While physics-based guidance can correct this, it faces two computational bottlenecks: expensive quantum-chemical evaluations (e.g., DFT) and the need to repeat such queries at every sampling step. We present Elign, a post-training framework that amortizes both costs. First, we replace expensive DFT evaluations with a faster, pretrained foundational machine-learning force field (MLFF) that estimates molecular energies and forces. Second, we eliminate repeated run-time queries by shifting physical steering to the post-training phase. To achieve the second amortization, we formulate reverse diffusion as a reinforcement learning problem and propose to use Group Relative Policy Optimization (GRPO) to fine-tune the denoising policy. Our objective combines a potential-based energy reward and a force-based stability reward, which are optimized in a disentangled fashion. Experiments show that Elign generates conformations with lower gold-standard DFT energies and forces, while improving stability. Crucially, inference remains as fast as unguided sampling, since no energy evaluations are required during generation.

---

[1]Department of Computer Science, Yale University, New Haven, USA [2]IQuestLab [3]Program in Computational Biology and Biomedical Informatics, Yale University, New Haven, USA. Correspondence to: Mark Gerstein <pi@gersteinlab.org>.

*Proceedings of the 43rd International Conference on Machine Learning*, Seoul, South Korea. PMLR 306, 2026. Copyright 2026 by the author(s).

## 1. Introduction

The generation of realistic three-dimensional molecular conformations is a central problem in computational chemistry, materials science, and drug discovery (Xu et al., 2023; Hoogeboom et al., 2022). For practical use, a generative model must satisfy two requirements. First, it must respect the symmetries of physics, most notably invariance to rigid body translations and rotations. Second, it must generate samples that correspond to low energy and physically stable configurations.

Score-based diffusion models with E(3)-equivariant architectures have shown notable performance in molecular generation (Xu et al., 2023; Hoogeboom et al., 2022; Cornet et al., 2025). In the idealized case where the training data are drawn from a true thermodynamic equilibrium, the data distribution itself would follow the Boltzmann law. Under this condition, maximizing likelihood naturally coincides with favoring mechanically plausible, low-energy configurations. In practice, however, this rarely holds. Standard datasets are typically constructed via semi-empirical relaxations (Bannwarth et al., 2019) or heuristic enumeration (Ramakrishnan et al., 2014; Isert et al., 2022), approximating equilibrium only coarsely. Consequently, the resulting model replicates the biases of the dataset generation rather than a true equilibrium distribution. This gap highlights a limitation of diffusion models: the likelihood objective does not, by design, inherently enforce physical stability.

A direct way to mitigate this is to introduce physical guidance or rewards during the generation process (Wu et al., 2022). In this setting, an oracle approximating the potential energy surface penalizes mechanically unstable configurations during sampling. While prior work (Zhou et al., 2025) has attempted to use first-principles oracles like density functional theory (DFT) (Hohenberg & Kohn, 1964; Kohn & Sham, 1965), the computational cost is often high. As a result, most approaches are forced to rely on tractable surrogates and compromises, such as applying rewards at terminal sampling stages (Zhou et al., 2025), relying on post-processing (Wu et al., 2022), or utilizing semi-empirical potential methods (Shen et al., 2024; Zhou et al., 2025). Notably, many guidance-based approaches (Shen et al., 2024) require *run-time alignment*, meaning that energy or force

evaluations must be performed during sampling, incurring additional computational overhead at inference. Another line of work reframes reward-guided diffusion as stochastic optimal control, leading to algorithms such as Adjoint Matching (Havens et al., 2025) and Adjoint Schrödinger Bridge Sampler (Liu et al., 2025a). These methods rely on adjoint equations that assume reward differentiability with respect to the state. In molecular diffusion models with mixed discrete and continuous variables, this assumption could fail, requiring gradients to be approximated using zeroth-order or surrogate estimators, e.g., Simultaneous Perturbation Stochastic Approximation (SPSA) (Spall, 1992; Shen et al., 2024).

MLFFs offer a natural mechanism for injecting physical constraints into generative models (Chmiela et al., 2017; Schütt et al., 2017). By amortizing the cost of expensive first-principles calculations (Car & Parrinello, 1985) into their training phase, inference reduces to efficient neural network evaluations of energies and forces (Behler, 2021; Unke et al., 2021). We see this as the *first level of amortization* in our framework. Moreover, a suggestive connection exists between MLFFs and diffusion models: the denoising objective used to train diffusion models (Hoogeboom et al., 2022; Xu et al., 2023) has also proven effective for MLFF pretraining (Zaidi et al., 2022; Feng et al., 2023). It is also worth noting that historically, MLFFs are typically trained on narrow, system-specific datasets (Chmiela et al., 2017; 2018) such as single molecules, small chemical families, or homogeneous materials. Thus, they do not define a unified potential across the vast chemical space spanned by modern diffusion models. Recent "foundation" MLFFs substantially address these limitations (Amin et al., 2025; Mannan et al., 2025; Unke et al., 2021; Wood et al., 2025): trained on large, chemically diverse quantum-mechanical datasets (Chanussot et al., 2021; Tran et al., 2023; Eastman et al., 2023; Unke et al., 2024), they approximate potential energy surfaces across wide molecular classes while retaining amortized efficiency.

MLFFs could enable physically grounded simulation pipelines, such as molecular dynamics or equilibrium sampling. They can also provide guidance that is more accurate than purely data-driven generators. However, those methods remain expensive when many diverse conformations must be generated, as it requires long trajectories and repeated force evaluations. This motivates a *second level of amortization*: compiling physical constraints into the diffusion model itself, shifting computation from inference to a post-training stage. By *post-training*, we mean an additional training stage applied after the likelihood-based pretraining. Together, MLFFs amortize quantum-mechanical calculations into reusable potentials, while post-training amortizes repetitive inference into the denoiser, eliminating run-time overhead. Based on these, we introduce Elign, a frame-work that fine-tunes E(3)-equivariant diffusion models using MLFF-derived rewards. Elign employs potential-based reward shaping derived from MLFF energies. Both energy and force signals are combined under a disentangled group-relative policy optimization scheme, analogous to MLFF training. Experiments on QM9 and GEOM-Drugs show that Elign generates low-energy, mechanically stable conformations, outperforming run-time-guided methods while matching the inference speed of unguided samplers.

## 2. Preliminaries

**Denoising diffusion models.** Diffusion models define a generative distribution $p_\theta(z_0)$ over data $p_{\text{data}}(z_0)$ on a space $\mathcal{Z}$ by learning to reverse an iterative forward noising process. The forward process is defined by the SDE $dz_t = b_t(z_t)\, dt + \sigma_t\, d\mathbf{B}_t$ with initial condition $z_0 \sim p_{\text{data}}$. Let $p_t = \text{Law}(z_t)$. For the general forward diffusion $dz_t = -\frac{1}{2}\beta_t z_t\, dt + \sqrt{\beta_t}\, d\mathbf{B}_t$, the transition distribution is $p(z_t|z_0) = \mathcal{N}(z_t; \alpha_t z_0, \bar{\sigma}_t^2 \mathbf{I})$ with $\alpha_t = \exp(-\frac{1}{2}\int_0^t \beta_s ds)$ and $\bar{\sigma}_t^2 = 1 - \exp(-\int_0^t \beta_s ds)$. With a constant noise schedule $\beta_t = 2$, we recover the Ornstein–Uhlenbeck forward process $dz_t = -z_t\, dt + \sqrt{2}\, d\mathbf{B}_t$ with $\alpha_t = e^{-t}$ and $\bar{\sigma}_t^2 = 1 - e^{-2t}$. This gives a signal-to-noise ratio $\text{SNR}(t) := \alpha_t^2/\bar{\sigma}_t^2$ that decreases monotonically in $t$, ensuring that as $t \to T$, $p_T \approx \mathcal{N}(0, \mathbf{I})$. To define the reverse process, let $z_t^{\leftarrow} = z_{T-t}$ denote the time-reversed trajectory with law $p_t^{\leftarrow} = p_{T-t}$. By Nelson's theorem (Nelson, 1967) with mild regularity conditions, the reverse-time process satisfies the drift relation $b_t(z) + b_{T-t}^{\leftarrow}(z) = a_t \nabla \log p_t(z)$ where $a_t = \sigma_t \sigma_t^{\top}$, yielding the reverse SDE $dz_t^{\leftarrow} = (-b_{T-t}(z_t^{\leftarrow}) + a_{T-t}\nabla \log p_{T-t}(z_t^{\leftarrow}))\, dt + \sigma_{T-t}\, d\mathbf{B}_t$. In practice, the score function is approximated by a neural network $s_\theta : \mathcal{Z} \times [0, T] \to \mathcal{Z}$ trained via denoising score matching by minimizing $\mathbb{E}\left[\|s_\theta(z_t, t) - \nabla_{z_t} \log p(z_t|z_0)\|^2\right]$.

**Equivariant diffusion models.** Let $\mathcal{G}$ be a group acting on a space $\mathcal{Z}$. A function $f : \mathcal{Z} \to \mathcal{Z}$ is $\mathcal{G}$-equivariant if $f(g \cdot z) = g \cdot f(z)$ for all $g \in \mathcal{G}, z \in \mathcal{Z}$, and a distribution $p$ on $\mathcal{Z}$ is $\mathcal{G}$-invariant if its density satisfies $p(g \cdot z) = p(z)$. To construct a diffusion model that generates samples from a $\mathcal{G}$-invariant law, both the forward and reverse SDEs must respect this symmetry. For the forward process, the Fokker–Planck equation $\partial_t p_t = \frac{1}{2}\langle \sigma_t \sigma_t^{\top}, \nabla^2 p_t \rangle - \nabla \cdot (b_t p_t)$ preserves invariance when $b_t$ is equivariant and $\sigma_t \sigma_t^{\top}$ commutes with the group action. By defining a $\mathcal{G}$-equivariant forward process, the true reverse-time drift is automatically $\mathcal{G}$-equivariant, since it is composed of an equivariant drift $b_t$ and an equivariant score function $\nabla \log p_t$ (for a $\mathcal{G}$-invariant distribution $p_t$, $\nabla \log p_t(g \cdot z) = g \cdot \nabla \log p_t(z)$). In practice, the unknown true score is estimated by a parameterized model $s_\theta(z_t, t)$. To preserve this symmetry, the score model must be parameterized as a $\mathcal{G}$-equivariant neural network.

The reverse process must also be initiated by drawing samples from a $\mathcal{G}$-invariant terminal distribution $p_T$.

**$N$-body data and subspace diffusion.** An $N$-body system is defined by state $\boldsymbol{z} = [\mathbf{x}, \mathbf{h}] \in \mathbb{R}^{N \times 3} \times \mathbb{R}^{N \times d_h}$, comprising positions $\mathbf{x} \in \mathcal{X}$ and invariant features $\mathbf{h} \in \mathcal{H}$ (e.g., atom types). The physical distribution $p(\boldsymbol{z})$ is invariant under the Euclidean group E(3) acting on $\mathcal{X}$. Translation invariance implies that $p(\boldsymbol{z})$ is not normalizable on the full space $\mathbb{R}^{3N}$. We therefore restrict the diffusion to the linear subspace of zero center-of-mass (CoM) configurations (Hoogeboom et al., 2022), $\mathcal{M} = \{\mathbf{x} \in \mathbb{R}^{N \times 3} \mid \sum_i \mathbf{x}_i = \mathbf{0}\}$. We define the projection operator $\mathbf{P}_{\text{CoM}}$ as a block-diagonal matrix: on positions it is the $3N \times 3N$ centering projector, while on features it acts as identity. Explicitly, $\mathbf{P}_{\text{CoM}} = \text{diag}(\mathbf{P}_{\mathcal{M}}, \mathbf{I}_{Nd_h})$ where $\mathbf{P}_{\mathcal{M}}$ projects onto $\mathcal{M}$. The forward SDE is modified to diffuse only within this subspace: $d\boldsymbol{z}_t = -\frac{1}{2}\beta_t \boldsymbol{z}_t\, dt + \sqrt{\beta_t}\,\mathbf{P}_{\text{CoM}}\, d\mathbf{B}_t$. Because $\mathbf{P}_{\mathcal{M}}$ is idempotent (that is, $\mathbf{P}_{\mathcal{M}} \circ \mathbf{P}_{\mathcal{M}} = \mathbf{P}_{\mathcal{M}}$) and commutes with the E(3) action restricted to rotations and reflections, the marginal distributions $p_t$ remain supported on $\mathcal{M}$ and retain E(3)-invariance. The score network $s_\theta$ is parameterized to be E(3)-equivariant and to output both coordinate and feature components, $s_\theta(\boldsymbol{z}_t, t) = [s_\theta^{(\mathbf{x})}(\boldsymbol{z}_t, t), s_\theta^{(\mathbf{h})}(\boldsymbol{z}_t, t)] \in \mathbb{R}^{N \times 3} \times \mathbb{R}^{N \times d_h}$. We enforce the zero-CoM constraint by projecting only the coordinate component (equivalently, applying $\mathbf{P}_{\mathcal{M}}$ to coordinate updates) at each denoising step.

**Machine-learning force fields.** Machine-learning force fields (MLFFs) serve as efficient surrogates for quantum-mechanical (QM) calculations, enabling dynamics simulations at scales inaccessible to *ab initio* methods. We assume access to a *quantum chemistry oracle* which, for any given molecular configuration $\boldsymbol{z}$, computes a reference energy $E^{\text{ref}}$ and reference atomic forces $\mathbf{F}^{\text{ref}}$. As direct queries to such oracles are computationally expensive, often scaling cubically with the number of atoms, MLFFs amortize this cost by training a neural network to approximate the potential energy function $E_\phi(\boldsymbol{z})$. Atomic forces are typically obtained via automatic differentiation as the negative gradient of this potential, $\mathbf{F}_\phi^{(i)} = -\nabla_{\mathbf{x}^{(i)}} E_\phi(\boldsymbol{z})$, though they may also be parameterized directly via a dedicated force regression head. In practice (e.g., (Wood et al., 2025)), one can first train a direct force-regression head and then fine-tune via automatic differentiation. During training, the network parameters $\phi$ are optimized by minimizing a combined energy-and-force matching loss over a dataset of reference calculations:

$$\mathcal{L} = \lambda_E \left(E_\phi(\boldsymbol{z}) - E^{\text{ref}}\right)^2 + \lambda_F \sum_{i=1}^{N} \left\|\mathbf{F}_\phi^{(i)}(\boldsymbol{z}) - \mathbf{F}^{\text{ref},(i)}(\boldsymbol{z})\right\|^2, \tag{1}$$

where $\lambda_E$ and $\lambda_F$ are weighting coefficients. Physical consistency imposes strict symmetry constraints: $E_\phi$ must be E(3)-invariant (unaffected by global rotation, translation, and inversion), while the derived forces must be E(3)-equivariant. These constraints are enforced via specialized invariant or equivariant network architectures. Recently,

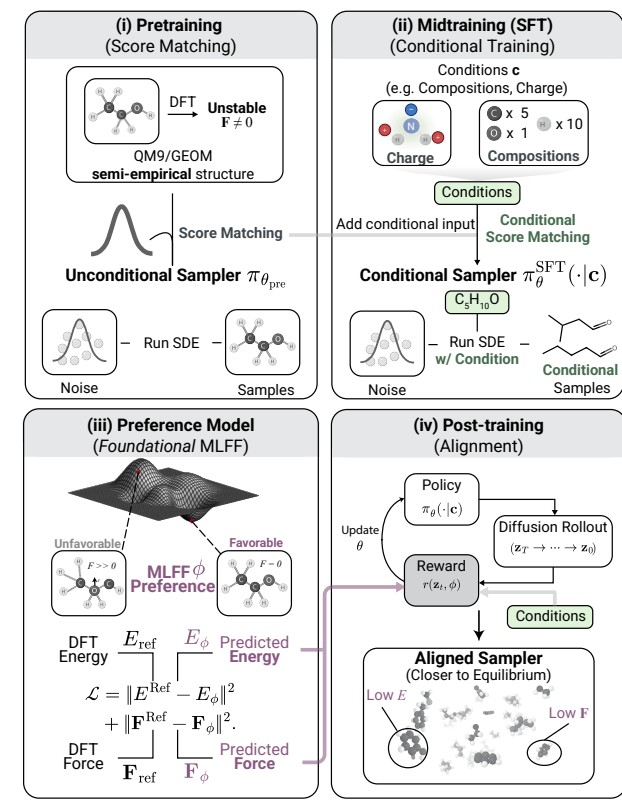

*Figure 1.* **Staged pipeline for equilibrium molecular generation.** (i) **Pretraining:** score-matching trains an E(3)-equivariant diffusion model $\pi_\theta^{\text{pre}}$ on approximate structures. (ii) **Optional SFT:** conditional fine-tuning improves adherence to discrete specifications. (iii) **Preference model:** a foundation MLFF $\phi$ provides energy/force signals. (iv) **Post-training:** RL fine-tunes the sampler to improve stability without run-time oracle calls.

foundational MLFFs have scaled this paradigm to massive heterogeneous datasets, learning universal potentials that are *transferable* across diverse chemical systems.

## 3. Diffusion post-training with a force-field alignment model

**Setup.** As shown in Figure 1, we organize equilibrium molecular generation into a staged pipeline analogous to LLM training: base model training, optional supervised conditioning, preference modeling, and post-training alignment. (i) **Pretraining (score matching):** We train an equivariant diffusion model $\pi_{\theta_{\text{pre}}}$ via score matching on large collections of approximate equilibrium structures (e.g., QM9-style semi-empirical pipelines). This stage learns a broad generative distribution but does not enforce physical stability. (ii) **Optional SFT (conditional training):** When explicit constraints are required, one could optionally fine-tune the model to condition on variables such as by training a conditional diffusion model $\pi_{\theta_{\text{sft}}}(\cdot \mid \mathbf{c})$ on paired data $(\mathbf{c}, \boldsymbol{z})$ using the same denoising objective. This step corresponds to instruction SFT in LLMs and improves constraint adherence

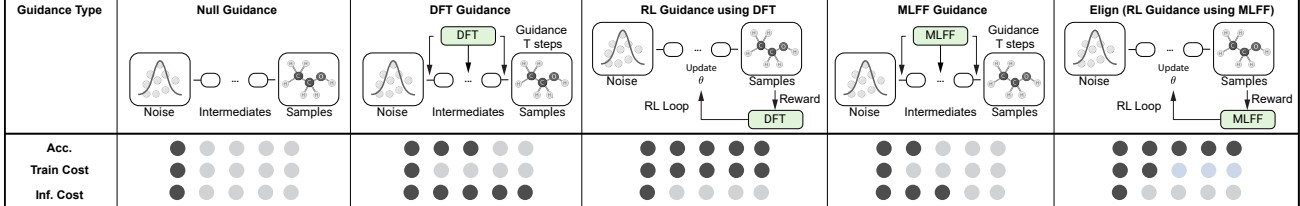

*Figure 2.* **Levels of amortization in physics-aware molecular generation.** (i) *Null Guidance*: unguided diffusion; (ii) *DFT Guidance*: per-step DFT queries (accurate but inference-prohibitive); (iii) *RL Guidance using DFT*: DFT as RL reward (amortized inference, still training-prohibitive); (iv) *MLFF Guidance*: MLFF replaces DFT per-step (first amortization); (v) *Elign*: MLFF rewards compiled into the policy at post-training so inference is unguided (second amortization). Light blue dots: cost paid by a third party that releases the foundation MLFF.

without introducing preference optimization[1]. (iii) **Preference model (foundation MLFF):** We use a pretrained foundation MLFF $\phi$ as a preference model, providing energies and forces that quantify thermodynamic stability and mechanical equilibrium. (iv) **Post-training (alignment):** We treat reverse diffusion as a trajectory-generating process and apply reinforcement learning to align $\pi_\theta$ with the MLFF-defined preferences, while limiting deviation from the pretrained or SFT-initialized model via trust-region regularization.

**Diffusion as an MDP.** Following Black et al. (2024), we formulate the iterative denoising process as a finite-horizon Markov Decision Process (MDP). We index reverse time with $t$ decreasing from $T$ to $0$. We initialize our method with a pretrained equivariant diffusion model, $\pi_{\theta_{\text{pre}}}$, which serves as the reference base policy. The state is defined by $S_t = (\mathbf{z}_t, t)$, comprising the latent molecular geometry $\mathbf{z}_t$ and the discrete reverse-diffusion time index $t$. The *policy* $\pi_\theta(\mathbf{z}_{t-1} \mid \mathbf{z}_t, t)$ executes a *one-step discretization* of the reverse SDE update (e.g., Euler–Maruyama). For simplicity, we assume first-order solvers so the unaugmented state $S_t$ is Markov. This induces a Gaussian policy parameterized by the score network: $\pi_\theta(\mathbf{z}_{t-1} \mid \mathbf{z}_t, t) = \mathcal{N}(\mathbf{z}_{t-1}; \mu_\theta(\mathbf{z}_t, t), \Sigma_t)$, where $\mu_\theta$ is the learned reverse mean and $\Sigma_t$ is given. The *action* is defined as a realization sampled from the Gaussian policy. It is important to distinguish the per-step policy, denoted $\pi(\mathbf{z}_{t-1} \mid \mathbf{z}_t, t)$, from the resulting distribution $\rho$, which arises from the sequential chaining of these policy steps. All stochasticity in the trajectory arises from the Gaussian perturbation inside the policy itself. This single source of randomness admits two equivalent interpretations: (i) as the noise term of the discretized reverse-time SDE, or (ii) as sampling from a stochastic policy. Both views refer to the same draw, since the policy itself is the SDE kernel. Here, we offer an alternative interpretation of this MDP through the lens of stochastic optimal control (of which maximum-entropy RL is a special case). In this view, the goal of RL is to learn a policy that acts as a "control knob," modulating the drift term $\mu_\theta(\cdot)$ of the

diffusion process to maximize the expected reward. This is also analogous to a *learnable guidance function* in diffusion guidance.

### 3.1. Alignment Objective

The objective of RL post-training is to ensure the final generated structures are physically valid. Validity requires two properties: thermodynamic stability (low potential energy) and mechanical equilibrium (small residual forces). To design the reward, we draw inspiration from training MLFFs, which utilize supervision from both energy and force labels to learn the potential energy surface (Eq. 1). Because force correlates with the gradient of energy, these objectives are coupled but distinct; a structure can have low energy but high unstable forces if it sits on a steep slope of the potential energy surface. We therefore define the *terminal alignment reward*, assigned only at the final step when $t = 0$. Specifically:

$$r_0^{(E)}(\mathbf{z}_0) := -E_\phi(\mathbf{z}_0); \; r_0^{(\mathbf{F})}(\mathbf{z}_0) := -\|\mathbf{F}_\phi(\mathbf{z}_0)\|_{\text{rms}}^2. \quad (2)$$

Here, $E_\phi$ and $\mathbf{F}_\phi$ are the energy and force predictions from the pretrained MLFF, and $\|\mathbf{F}\|_{\text{rms}}$ denotes the per-atom RMS force magnitude (i.e., $\|\mathbf{F}\|_{\text{rms}} := \frac{1}{\sqrt{N}}\|\mathbf{F}\|_F$).

**Energy-based potential shaping.** Optimizing solely for a terminal alignment reward yields a sparse learning signal over long reverse diffusion horizons. To facilitate credit assignment while preserving the optimal policy of the original terminal-reward MDP, we add an intermediate signal using *potential-based reward shaping* (PBRS). Traditionally, PBRS relies on a potential function $\Psi(S)$ that estimates the "closeness" of a current state to the goal, much like an admissible heuristic in A* search (Hart et al., 1968). In our setting, this role is naturally played by physics: because our objective is thermodynamic stability, the physical potential energy $E_\phi$ directly quantifies the distance from equilibrium.

The MLFF energy $E_\phi(\cdot)$ is physically meaningful only on clean atomic coordinates, not on the noisy intermediates $\mathbf{z}_t$. At each reverse step $t$ we therefore reconstruct a *predicted clean geometry* $\mathbf{z}_{0|t}$ from $\mathbf{z}_t$ using the diffusion posterior

---

[1]Due to the page limit, we include results on conditional generation in Appendix E.2.

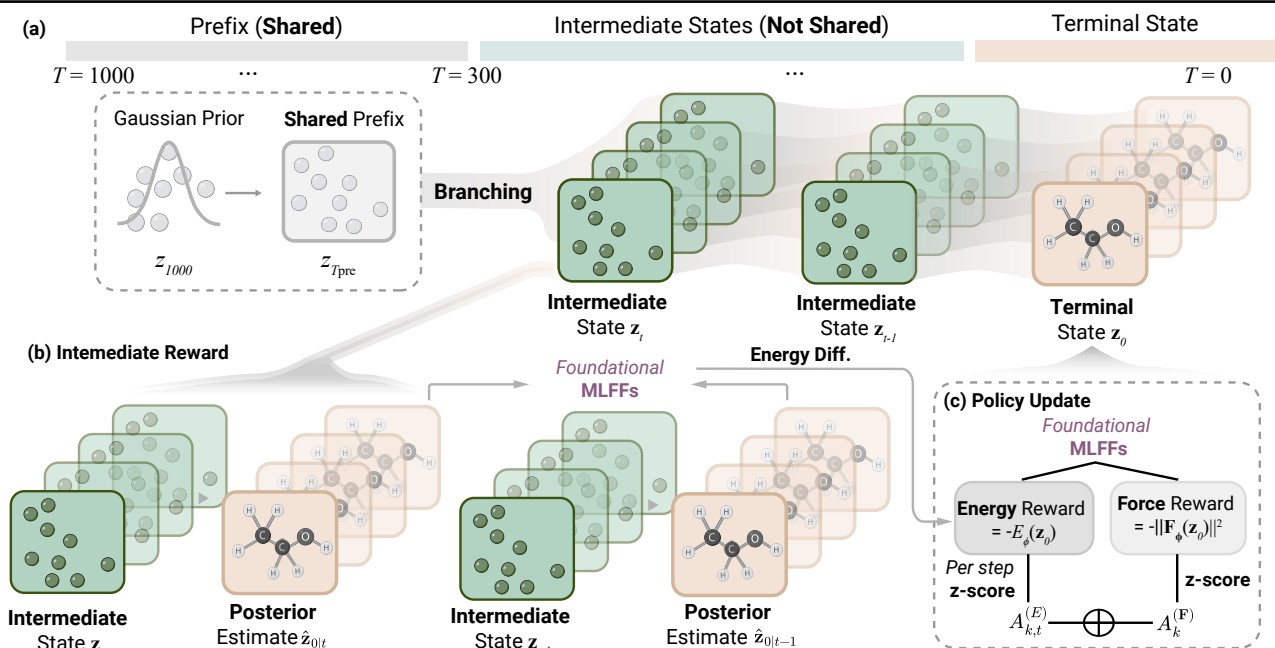

*Figure 3.* Overview of Elign. **(a)** Rollout branching with shared prefix: starting from a CoM-free Gaussian prior at $t = T$, an EGNN policy denoises to $t = 0$. At $t = T_{\text{prefix}}$, we cache a shared prefix state and branch into $K$ rollouts with independent noise. Each trajectory is propagated to its terminal state $z_0$ and scored by a foundational MLFF. **(b)** Energy-based reward shaping: intermediate predicted clean geometries $\hat{z}_{0|t}$ are evaluated by the MLFF, and local energy differences $E_\phi(\hat{z}_{0|t}) - E_\phi(\hat{z}_{0|t-1})$ provide dense shaping signals that bias the policy toward lower-energy conformations. **(c)** Disentangled GRPO: terminal energy and RMS force rewards are z-score normalized separately per timestep, then combined to compute the final advantage for policy updates.

mean estimate: $\hat{z}_{0|t} = \frac{1}{\alpha_t}(z_t + \bar{\sigma}_t^2 s_\theta(z_t, t))$. We define a shaping potential on this reconstructed estimate: $\Psi(S_t) := -E_\phi(\hat{z}_{0|t})$, $S_t = (z_t, t)$. The intermediate shaping reward for a transition $S_t \to S_{t-1}$ is then

$$r_t^{\text{shape}} := \gamma \, \Psi(S_{t-1}) - \Psi(S_t), \qquad t = T_{\text{prefix}}, \dots, 1. \tag{3}$$

This is the canonical PBRS form; for a fixed shaping potential $\Psi$, it preserves the set of optimal policies under the discounted return (Ng et al., 1999). In particular, if we define the (discounted) shaped return-to-go from state $S_t$ as $G_t^{\text{shape}} := \sum_{u=1}^{t} \gamma^{t-u} r_u^{\text{shape}}$, then the shaping contribution telescopes:

$$G_t^{\text{shape}} = \sum_{u=1}^{t} \gamma^{t-u} (\gamma \Psi(S_{u-1}) - \Psi(S_u)) = \gamma^t \Psi(S_0) - \Psi(S_t). \tag{4}$$

Intuitively, the return-to-go is the discounted difference between terminal and current state potentials. It is worth noting that in our implementation, $\Psi(S_t)$ is evaluated on a plug-in estimate $\hat{z}_{0|t}$ produced by the current denoiser, so PBRS should be viewed as a practical heuristic rather than a strict policy-invariance guarantee.

### 3.2. Theoretical View of the Alignment Objective

The rewards above specify what properties we desire in the final sample. We now adopt a distributional perspective to characterize the *terminal law* that post-training encourages. Because the diffusion policy is parameterized by an E(3)-equivariant architecture, post-training preserves invariance

of the terminal distribution. For analytical clarity, we focus on the energy objective and ignore PBRS.

Let $\mathcal{Z}$ denote the (CoM-free) configuration space of clean molecular structures, and identify the terminal denoised sample $z_0 \in \mathcal{Z}$. Let $\rho_{\theta_{\text{pre}}} \in \mathcal{P}(\mathcal{Z})$ denote the terminal distribution induced by the pretrained diffusion model $\pi_{\theta_{\text{pre}}}$, i.e., $z_0 \sim \rho_{\theta_{\text{pre}}}$ when sampling from $\pi_{\theta_{\text{pre}}}$. Any fine-tuned policy $\pi_\theta$ induces its own terminal law $\rho_\theta$ over $\mathcal{Z}$.

**Theorem 1** (Energy-aligned terminal distribution; Levine, 2018; Rafailov et al., 2023). *Under mild realizability of the policy class (full statement in App. 3), the trust-region regularized objective $\mathcal{J}(\rho) := \mathbb{E}_{z_0 \sim \rho}[-E_\phi(z_0)] - w_{\text{KL}} \, \text{KL}(\rho \| \rho_{\theta_{\text{pre}}})$ admits a unique maximizer*

$$\rho^\star(z) = \frac{1}{Z_\phi} \rho_{\theta_{\text{pre}}}(z) \exp(-\beta_{\text{eff}} \, E_\phi(z)), \quad \beta_{\text{eff}} := 1/w_{\text{KL}}, \tag{5}$$

*with normalizer $Z_\phi$.*

The aligned law $\rho^\star$ does *not* recover the exact Boltzmann distribution $\propto \exp(-\beta E_\phi)$; instead, it is the pretrained law $\rho_{\theta_{\text{pre}}}$ tilted by the MLFF energy, which moves the sampler closer to thermodynamic equilibrium while staying on the learned data manifold. Equivalently, defining an implicit "prior energy" $E_{\text{prior}}(z) := -\beta_{\text{eff}}^{-1} \log \rho_{\theta_{\text{pre}}}(z)$ gives $\rho^\star \propto \exp(-\beta_{\text{eff}}[E_\phi + E_{\text{prior}}])$, so the pretrained model acts both as a regularizer and as a support constraint. Post-training can thus be viewed as amortized sampling: instead of applying energy guidance at inference time via

repeated $E_\phi$ evaluations, RL *compiles* the corresponding Gibbs reweighting into the reverse diffusion policy. Next, we relate MLFF approximation error to distributional distance.

**MLFF approximation and distribution error.** Let $E_\star :$ $\mathcal{Z} \to \mathbb{R}$ be a target potential energy and $E_\phi$ its MLFF approximation. Fix a reference terminal law $\rho_{\theta_{\text{pre}}}$ and define $\rho_\star^\star(\boldsymbol{z}) \propto \rho_{\theta_{\text{pre}}}(\boldsymbol{z}) e^{-\beta_{\text{eff}} E_\star(\boldsymbol{z})}$, $\rho_\phi^\star(\boldsymbol{z}) \propto \rho_{\theta_{\text{pre}}}(\boldsymbol{z}) e^{-\beta_{\text{eff}} E_\phi(\boldsymbol{z})}$. We characterize the distribution error using the total variation distance:

**Theorem 2** (Energy error implies distribution error)**.** *Assume the MLFF uniformly approximates the target energy,* $\sup_{\boldsymbol{z} \in \mathcal{Z}} |E_\phi(\boldsymbol{z}) - E_\star(\boldsymbol{z})| \leq \delta$ *for some* $\delta \geq 0$. *Then the aligned terminal laws satisfy*

$$\|\rho_\phi^\star - \rho_\star^\star\|_{\text{TV}} \leq \tanh(\beta_{\text{eff}} \delta). \quad (6)$$

A key feature of Eq. (6) is that the right-hand side is *uniformly bounded by one* via $\tanh$, so terminal-law error remains controlled across all energy-approximation regimes rather than growing without bound. In the small-error or moderate-temperature regime $\tanh(\beta_{\text{eff}} \delta) \approx \beta_{\text{eff}} \delta$, so MLFF improvements translate *linearly* into closer Gibbs alignment; in the high-$\beta_{\text{eff}}$ regime, the bound saturates, letting the practitioner trade off MLFF accuracy against trust-region strength gracefully. The bound is worst-case in $\delta$; in practice we expect tighter behavior on the high-density region of the pretrained manifold, where the MLFF is more accurate. Next, we describe how this objective is implemented via reinforcement learning.

### 3.3. RL Training

**Force–energy disentangled GRPO.** To fine-tune the diffusion model, we require a policy gradient algorithm that is both sample-efficient and computationally inexpensive. Standard actor–critic methods are costly, as they would require training a separate value function alongside the score model. Moreover, in our setting the value function must be E(3)-invariant and defined over high-dimensional molecular states; learning such a critic via bootstrapping can introduce instability, in addition to extra compute. We instead adopt *Group Relative Policy Optimization* (GRPO) (Shao et al., 2024), a critic-free algorithm, and introduce a *disentangled* advantage formulation that separately accounts for energy and force signals.

*Shared-prefix grouping.* We first define a *group* over a diffusion trajectory. In large language models, all members of a group share the same prompt; for example, given a fixed math question, multiple responses are sampled in parallel and compared to identify higher-quality answers. Analogously, we extend this idea to diffusion by rolling out a common reverse-diffusion prefix from $T \to T_{\text{prefix}}$ (for

a chosen prefix time $T_{\text{prefix}} \in \{1, \ldots, T\}$) and caching $\boldsymbol{z}_{T_{\text{prefix}}}$. From this shared state we branch into $K$ stochastic continuations with fresh noise. This shared initialization has also been used in concurrent work (e.g., FlowGRPO (Liu et al., 2025b)) and lowers variance by making early denoising dynamics comparable.

*Two-channel advantages.* For each branched rollout $k$, we distinguish between dense energy feedback and sparse force feedback: (i) **Step-wise energy advantage:** For energy, we utilize the dense PBRS signal. We first compute the raw *return-to-go* at each step $t$: $G_{k,t}^{(E)} = \sum_{u=1}^{t} \gamma^{t-u} r_{k,u}^{E,\text{shape}}$. Crucially, we do **not** normalize $r_{k,u}$ per step; we first sum raw shaped rewards to preserve telescoping (Eq. (4)), then normalize only at the advantage level. (ii) **Trajectory-level force advantage:** For force, we assign a single scalar return to the entire trajectory: $G_k^{(\mathbf{F})} = - \left\| \mathbf{F}_\phi(\boldsymbol{z}_0^{(k)}) \right\|_{\text{rms}}^2$.

*Disentangled group-relative normalization.* Directly summing the raw rewards is undesirable, as their differing scales and temporal structures can lead to interference, with one signal dominating the optimization dynamics. We therefore standardize the two channels differently to mitigate these effects and to disentangle their contributions according to granularity. For **force**, we compute a single group-level mean $\mu_F$ and standard deviation $\sigma_F$. For **energy**, we compute statistics **per time-step** $t$, denoted $(\mu_{E,t}, \sigma_{E,t})$, across the $K$ rollouts. This ensures that the advantage at step $t$ is relative to the group's performance *at that specific step*. The total advantage for rollout $k$ at step $t$ is:

$$\hat{A}_{k,t} = w_E \underbrace{\frac{G_{k,t}^E - \mu_{E,t}}{\sigma_{E,t} + \eta}}_{\text{Step-wise Energy Adv.}} + w_{\mathbf{F}} \underbrace{\frac{G_k^F - \mu_F}{\sigma_F + \eta}}_{\text{Global Force Adv.}}. \quad (7)$$

Here, $\eta > 0$ is a small constant added for numerical stability, and the force term is broadcast to the entire trajectory.

*Clipped GRPO objective.* Let the probability ratio be $\xi_{k,t}(\theta) = \frac{\pi_\theta(\boldsymbol{z}_{t-1}^{(k)} | \boldsymbol{z}_t^{(k)}, t)}{\pi_{\theta_{\text{old}}}(\boldsymbol{z}_{t-1}^{(k)} | \boldsymbol{z}_t^{(k)}, t)}$, where all Gaussian log-probabilities are computed in the CoM-free subspace so the transition kernel has a proper density in $\mathbb{R}^{3N-3}$. The clipped surrogate objective aggregates over all steps using the per-step advantage $\hat{A}_{k,t}$: $L(\theta) = \mathbb{E}_k [\sum_{t=1}^{T_{\text{prefix}}} \min(\xi_{k,t}(\theta) \hat{A}_{k,t}, \text{clip}(\xi_{k,t}(\theta), 1-\varepsilon, 1+\varepsilon) \hat{A}_{k,t})]$, This objective maximizes (or minimizes) the likelihood of denoising trajectories proportional to their physics-based advantage, with clipping enforcing a *local* (per-step) trust region that stabilizes training; a *global* trust region can also be added via a KL penalty against the pretrained policy $\pi_{\theta_{\text{pre}}}$. Concretely, this is given as: $\mathcal{L}_{\text{reg}}(\theta) := \mathcal{L}(\theta) - w_{\text{KL}} \text{KL}(\pi_\theta \| \pi_{\theta_{\text{pre}}})$, where $w_{\text{KL}} > 0$ controls the strength of the global trust region. Note that this is a tractable proxy for terminal-law trust region defined in

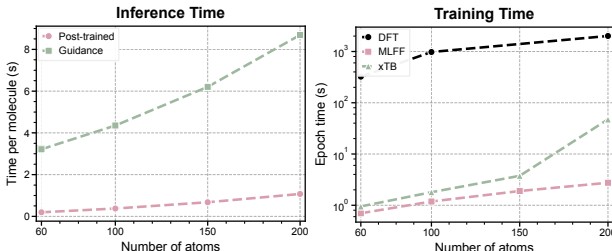

*Figure 4.* Computational efficiency comparison. **(Left)** Inference time per molecule for post-trained models versus guidance-based methods. **(Right)** Training epoch time comparing different reward oracles on a log scale. For a fair comparison, all methods use terminal-only rewards (no PBRS). Evaluation was conducted on a node with an NVIDIA H100 GPU and Intel Xeon Sapphire Rapids CPUs (Xeon Platinum 6458Q, AVX-512 enabled).

Sec. 3.2. We summarize our GRPO procedure in Algs. 1–2.

---

**Algorithm 1** GRPO post-training

---

1: **Input:** $\pi_{\theta_{\text{old}}}$, MLFF $\phi$, $T_{\text{prefix}}$, $K$, $(w_E, w_{\mathbf{F}})$, $\gamma$, clip $\varepsilon$.
2: **for** each iteration **do**
 3: Rollout $T \to T_{\text{prefix}}$ with $\pi_{\theta_{\text{old}}}$; cache $\mathbf{z}_{\text{start}}$. ▷ *Shared.*
 4: **for** $k = 1, \dots, K$ **do**
  5: $\tau_k \leftarrow \text{ROLLOUT}(\mathbf{z}_{\text{start}}, \pi_{\theta_{\text{old}}})$
  6: $(\{G_{k,t}^{(E)}\}_{t=1}^{T_{\text{prefix}}}, G_k^{(\mathbf{F})}) \leftarrow \text{REWARD}(\tau_k, \phi, \pi_{\theta_{\text{old}}})$.
 7: $\hat{A}_k^{(\mathbf{F})} \leftarrow (G_k^{(\mathbf{F})} - \mu_F)/(\sigma_F + \eta)$ over $k$.
 8: $\hat{A}_{k,t}^{(E)} \leftarrow (G_{k,t}^{(E)} - \mu_{E,t})/(\sigma_{E,t} + \eta)$ over $k$ for each $t$.
 9: $\hat{A}_{k,t} \leftarrow w_E \hat{A}_{k,t}^{(E)} + w_{\mathbf{F}} \hat{A}_k^{(\mathbf{F})}$;
 10: $\xi_{k,t} \leftarrow \pi_\theta(\mathbf{z}_{t-1}^{(k)} | \mathbf{z}_t^{(k)}, t) / \pi_{\theta_{\text{old}}}(\mathbf{z}_{t-1}^{(k)} | \mathbf{z}_t^{(k)}, t)$.
 11: Update $\theta$ with the GRPO objective using $(\xi_{k,t}, \hat{A}_{k,t})$; ▷ *Optionally add a KL Term.*
 12: set $\theta_{\text{old}} \leftarrow \theta$.

---

**Algorithm 2** REWARD: energy PBRS return-to-go + terminal force

---

1: **function** REWARD($\tau, \phi, \pi_{\theta_{\text{old}}}$)
 2: Extract $\{\mathbf{z}_t\}_{t=0}^{T_{\text{prefix}}}$ from $\tau$.
 3: **for** $t = T_{\text{prefix}}, \dots, 0$ **do**
  4: Compute $\hat{\mathbf{z}}_{0|t}$ from $\mathbf{z}_t$ using $\pi_{\theta_{\text{old}}}$. ▷ *Stop Grad.*
  5: $\Psi_t \leftarrow -E_\phi(\hat{\mathbf{z}}_{0|t})$.
 6: **for** $t = 1, \dots, T_{\text{prefix}}$ **do**
  7: $G_t^{(E)} \leftarrow \gamma^t \Psi_0 - \Psi_t$. ▷ *Eq. (4)*
 8: $G^{(\mathbf{F})} \leftarrow -\|\mathbf{F}_\phi(\mathbf{z}_0)\|_{\text{rms}}^2$.
 9:
 10: **return** $\{G_t^{(E)}\}_{t=1}^{T_{\text{prefix}}}, G^{(\mathbf{F})}$.

---

# 4. Results

**Implementation.** We use the public EDM (Hoogeboom et al., 2022) checkpoint as our pretrained base model and UMA (Wood et al., 2025) as the preference model. UMA is a machine-learning force field trained on OMol25 (Levine et al., 2025), OC20 (Chanussot et al., 2021), ODAC23 (Sriram et al., 2024), OMat24 (Barroso-Luque et al., 2024), etc., that predicts per-molecule energies and per-atom forces from atom types and 3D coordinates. UMA has two vari-

ants: UMA-1p1-S and UMA-1p1-M. Unless stated otherwise, Elign uses UMA-1p1-M with potential-based reward shaping (PBRS). We evaluate on QM9 (Ramakrishnan et al., 2014) and GEOM-Drugs (Axelrod & Gómez-Bombarelli, 2022). QM9 contains 130k small molecules with up to 9 heavy atoms (29 atoms including hydrogens). GEOM-Drugs consists of larger organic compounds with up to 181 atoms (44.2 on average) across 37 million conformations for around 450k molecules. Following Xu et al. (2023), we report atom stability (A), molecule stability (M), RDKit validity (V), and Validity×Uniqueness (V×U). QM9 results average three runs of 10,000 samples; GEOM-Drugs uses 1,024 samples. For GEOM-Drugs, we omit M and V×U due to limitations in ground-truth bond inference (bond inference is unreliable for larger molecules with 60–200 atoms) and near-saturated uniqueness ($\approx 100\%$). This is in line with community standards (Xu et al., 2023). Baseline results are taken from published work whenever possible. Although UMA's training data may overlap with QM9 or GEOM-Drugs, the 3D conformations generated by EDM are novel. We use UMA solely as a frozen reward oracle, and our metrics (e.g. RDKit validity, DFT energies/forces) are computed independently of UMA.

**QM9.** Table 1 reports the benchmark results on QM9. Elign improves molecule stability from 82.00% (EDM) to 93.70% and increases V×U from 90.70% to 95.31%. Compared with DFT-based RL (EDM+DFT), Elign matches atom stability and achieves higher V×U (95.31% vs. 92.87%) without DFT queries during training or sampling.

**GEOM-Drugs.** Table 3 evaluates larger drug-like molecules. Elign reaches 87.94% atom stability and 99.40% validity, improving over EDM (81.3% / 91.9%) and over RLPF (xTB rewards) on atom stability (87.94% vs. 87.52%).

**Transfer across backbone, reward, and sampler family.** To test whether the gains are tied to the EDM + UMA pairing, we run three controlled transfer experiments on GEOM-Drugs (Table 4). *Backbone transfer:* replacing EDM with GeoLDM (Xu et al., 2023) while keeping UMA-1p1-M as the reward improves atom stability from 84.4% to 88.92%. *Reward-model transfer:* keeping EDM and replacing UMA with the smaller MACE-POLAR force field (Batatia et al., 2026) still improves atom stability from 81.3% to 86.98%. *Sampler-family transfer:* swapping diffusion for a flow-matching generator (EquiFM (Song et al., 2023)) yields 87.25% atom stability, up from 84.1%. Together, these results indicate that Elign acts as a portable post-training alignment layer rather than an EDM + UMA-specific recipe.

*Flow-matching adaptation.* Extending Elign to a flow-based generator requires converting the deterministic ODE into a stochastic policy over the tuned suffix. Following Liu et al. (2025b), we use the marginal-preserving SDE $d\mathbf{z}_t =$

$(v_\theta(\boldsymbol{z}_t, t) - \frac{\sigma_t^2}{2}\nabla \log p_t(\boldsymbol{z}_t))\mathrm{d}t + \sigma_t\,\mathrm{d}w$. For rectified flow the score admits a closed form in the velocity field, and Euler–Maruyama discretization gives the Gaussian policy

$$\boldsymbol{z}_{t+\Delta t} = \boldsymbol{z}_t + \Big[v_\theta(\boldsymbol{z}_t, t) + \frac{\sigma_t^2}{2t}(\boldsymbol{z}_t + (1-t)\,v_\theta(\boldsymbol{z}_t, t))\Big]\Delta t$$
$$+ \sigma_t\sqrt{\Delta t}\,\epsilon,$$

(8)

whose log-probabilities are available in closed form for GRPO. The shared prefix $(T \to T_{\mathrm{prefix}})$ is rolled out with the original deterministic ODE; only in the suffix $(T_{\mathrm{prefix}} \to 0)$ do we use the stochastic form. At inference we set $\sigma_t = 0$ and recover the deterministic flow sampler end-to-end.

**Alanine dipeptide (finite-temperature equilibrium).** To test the framework beyond minimum-energy conformer benchmarks, we evaluate Elign on alanine dipeptide, a biomolecular benchmark whose Ramachandran $(\phi, \psi)$ landscape has multiple metastable basins separated by free-energy barriers and is a sensitive probe of finite-temperature distributional fidelity. We pretrain a fixed-topology unconditional EDM-style generator on 300 K MD trajectories, then post-train it with our GRPO procedure using a UMA-1p1-M energy reward, using a trajectory-level split: 36k/4k train/val frames from one MD trajectory and 20k frames from an independent held-out trajectory for evaluation. On 256 generated samples, Elign reduces $(\phi, \psi)$-JSD from $0.6432$ to $\mathbf{0.5078}$ and energy $W_1$ from $0.831$ to $\mathbf{0.254}$ ($3.3\times$ smaller), outperforming run-time energy guidance (Table 5). We note Elign is not an exact de novo Boltzmann sampler: Theorem 1 shows the aligned terminal law is a Gibbs tilt of the pretrained law; closing the residual gap would require correcting for the pretrained model's implicit prior energy.

**Ablation analysis.** Table 2 reports controlled ablations on QM9. Our default configuration achieves the strongest V×U (95.31%). Replacing dense PBRS with sparse terminal rewards reduces V×U to 93.85%, indicating that intermediate reward signals improve credit assignment over long diffusion horizons. Substituting UMA-1p1-M with the smaller UMA-1p1-S exposes a stability–diversity tradeoff: molecule stability slightly improves (94.83% vs. 93.70%) but V×U drops 4.5 points, plausibly because the smaller model produces higher-variance gradients that concentrate samples around fewer local minima. Decomposing the reward, energy-only rewards yield low molecule stability (86.00%) while force-only rewards reach high stability but low diversity (V×U = 90.21%); jointly optimizing both with dense shaping balances the two.

**DFT Oracle Evaluation.** To verify that Elign's improvements reflect genuine physical quality rather than exploiting the MLFF reward, we evaluate generated molecules using independent DFT calculations. Specifically, we use PySCF (Sun et al., 2020) with B3LYP/6-31G(d) to compute total energies and nuclear gradients, then report formation energy

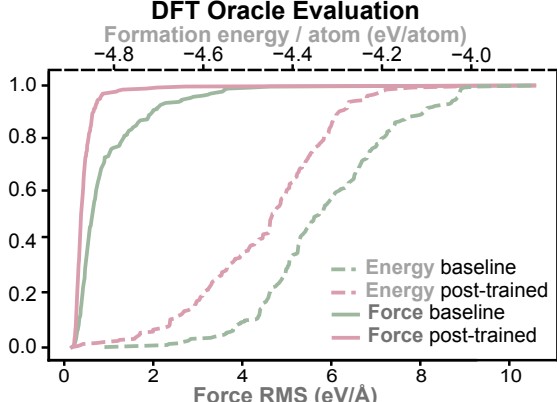

*Figure 5.* DFT oracle evaluation of generated molecules from QM9. Cumulative distribution functions comparing baseline and post-trained models on force RMS (left) and formation energy per atom (right). The results are obtained from 1024 samples.

*Table 1.* QM9 3D generation. Atom stability (A), molecule stability (M), RDKit validity (V), and Validity×Uniqueness (V×U).

| Model | A [%] ↑ | M [%] ↑ | V [%] ↑ | V×U [%] ↑ |
|---|---|---|---|---|
| EDM (Hoogeboom et al., 2022) | 98.70 | 82.00 | 91.90 | 90.70 |
| EDM-BRIDGE (Wu et al., 2022) | 98.80 | 84.60 | 92.00 | 90.70 |
| GeoLDM (Xu et al., 2023) | 98.90 | 89.40 | 93.80 | 92.70 |
| EDN (Cornet et al., 2025) | 98.90 | 89.10 | 94.80 | 92.60 |
| UniGEM (Feng et al., 2024) | 99.00 | 89.80 | 95.00 | 93.20 |
| GeoBFN (Song et al., 2024) | 99.08 | 90.87 | 95.31 | 92.96 |
| RLPF (EDM + DFT PPO) (Zhou et al., 2025) | 99.08 | 93.37 | 98.22 | 92.87 |
| EDM + Rejection Sampling (Zhou et al., 2025) | 98.99 | 89.47 | 93.20 | 92.60 |
| GeoLDM + xTB Guidance (Shen et al., 2024) | 99.02 | 90.60 | 91.40 | 91.40 |
| EDM + Soft Metropolis–Hastings (Feng et al., 2025) | **99.56** | 91.70 | **98.70** | 83.20 |
| GeoLDM + Soft Metropolis–Hastings (Feng et al., 2025) | 99.09 | 91.30 | 95.10 | 94.90 |
| EDM + UMA-1p1-S Guidance | 98.90 | 87.00 | 92.54 | 92.34 |
| EDM + UMA-1p1-M Guidance | 98.94 | 87.25 | 92.98 | 92.98 |
| Elign (EDM + UMA-1p1-M) | 99.33 | 93.70 | 98.32 | 95.31 |
| Data (Ground Truth) | 99.00 | 95.20 | 97.70 | 97.70 |

per atom (total energy minus isolated atom energies, divided by atom count). Figure 5 shows cumulative distributions for baseline EDM and Elign-trained models. Post-training shifts both distributions toward improved values: reduced force RMS indicates conformations closer to stationary points, while lower formation energy reflects increased thermodynamic stability. Because these metrics come from a DFT oracle not used during training, the results confirm that Elign improves physical fidelity beyond standard stability metrics.

**Speed.** Figure 4 reports wall-clock scaling. DFT evaluation is orders of magnitude slower than MLFF/UMA and scales steeply with system size; xTB also becomes expensive beyond $\sim$150–200 atoms. This observation supports the first level of amortization in our framework. During inference, run-time guidance is slower because it requires oracle gradients at each denoising step, whereas Elign uses the unguided EDM sampling procedure; across 60–200 atoms, post-trained sampling is about 8–16× faster in our benchmark. This supports the second level of amortization.

## Conformer Quality Comparison

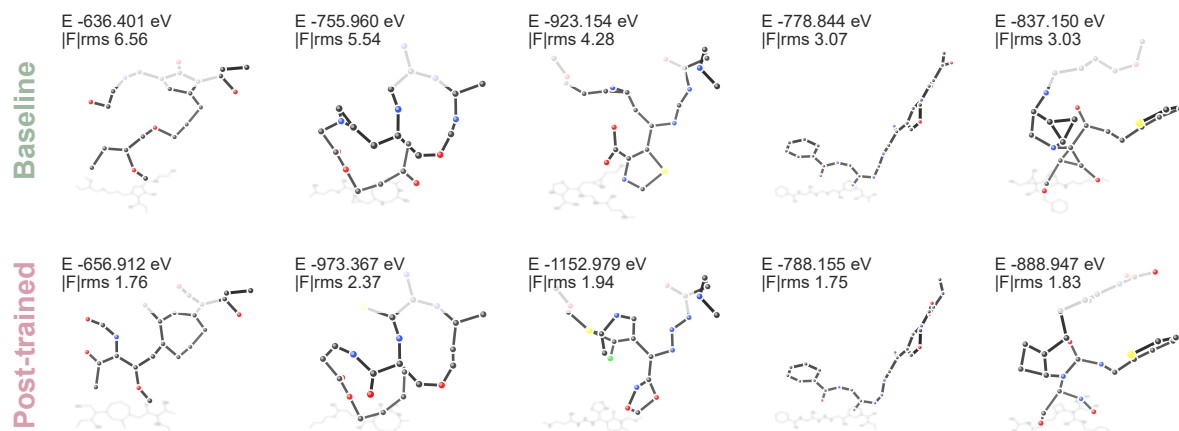

*Figure 6.* **Conformer quality before vs. after Elign post-training.** Representative conformers from the pretrained baseline (top row) and the Elign post-trained model (bottom row), annotated with MLFF-predicted energy ($E$) and RMS force norm. Post-training consistently lowers both $E$ and residual force, indicating geometries closer to local equilibrium.

*Table 2.* QM9 ablations. We vary reward composition (E vs F), reward density (Sparse terminal vs Dense PBRS shaping), and MLFF capacity (UMA-1p1-S vs UMA-1p1-M).

| Reward | | | MLFF | | | | | |
|---|---|---|---|---|---|---|---|---|
| E | F | PBRS | S | M | A [%]↑ | M [%]↑ | V [%]↑ | V×U [%]↑ |
| ✓ | ✓ | ✓ | | ✓ | 99.33 | 93.70 | **98.32** | **95.31** |
| ✓ | ✓ | ✓ | ✓ | | **99.42** | 94.83 | 97.53 | 90.81 |
| ✓ | ✓ | | | ✓ | 99.02 | 93.75 | 96.54 | 93.85 |
| ✓ | | | | ✓ | 98.70 | 86.00 | 91.70 | 91.70 |
| | ✓ | | | ✓ | 99.40 | **94.92** | 96.84 | 90.21 |

*Table 3.* GEOM-drug 3D generation. Atom stability (A) and RDKit validity (V).

| Model | A [%]↑ | V [%]↑ |
|---|---|---|
| EDM (Hoogeboom et al., 2022) | 81.3 | 91.9 |
| EDM-BRIDGE (Wu et al., 2022) | 82.4 | 91.9 |
| GeoLDM (Xu et al., 2023) | 84.4 | 99.3 |
| EDN (Cornet et al., 2025) | 87.0 | 92.9 |
| UniGEM (Feng et al., 2024) | 85.1 | 98.4 |
| GeoBFN (Song et al., 2024) | 85.6 | 92.08 |
| RLPF (EDM+xTB (Zhou et al., 2025) PPO) | 87.52 | 99.20 |
| Elign (EDM+ UMA-1p1-M) | **87.94** | **99.40** |
| Data (Ground Truth) | 86.5 | 99.9 |

*Table 4.* Transfer of Elign across generator backbones, MLFF reward models, and sampler families on GEOM-Drugs. Atom stability before/after post-training; the diffusion+UMA pairing is the original Elign configuration.

| Transfer axis | Backbone | Reward MLFF | Sampler | Before [%] | After [%] |
|---|---|---|---|---|---|
| Original (reference) | EDM | UMA-1p1-M | Diffusion | 81.3 | **87.94** |
| Backbone transfer | GeoLDM | UMA-1p1-M | Diffusion | 84.4 | **88.92** |
| Reward-model transfer | EDM | MACE-POLAR (Batatia et al., 2026) | Diffusion | 81.3 | **86.98** |
| Sampler-family transfer | Flow Matching | UMA-1p1-M | Flow | 84.1 | **87.25** |

*Table 5.* Alanine dipeptide at 300 K. Held-out evaluation against an independent MD trajectory using 256 generated samples. Lower is better for both metrics.

| Metric | Pretrained | +Guid. | +Elign |
|---|---|---|---|
| $(\phi, \psi)$-JSD ↓ | 0.6432 | 0.6231 | **0.5078** |
| Energy $W_1$ ↓ | 0.831 | 0.782 | **0.254** |

et al., 2023; 2024; Feng et al., 2024; Qiang et al., 2023). Extensions incorporating physical guidance or control include energy-informed priors (Wu et al., 2022; Shen et al., 2024), reinforcement learning with DFT (Zhou et al., 2025), and stochastic control formulations such as Adjoint Matching and Adjoint Schrödinger Bridge Samplers (Havens et al., 2025; Liu et al., 2025a). While effective in biasing samples toward desired properties, these methods typically rely on sparse rewards, differentiable objectives, or run-time oracle evaluations, potentially limiting their scalability. We also include an expanded related-work appendix.

## 6. Conclusion & Outlook

We presented Elign, a force–energy disentangled GRPO post-training framework that aligns equivariant molecular diffusion with foundational MLFF preferences, improving thermodynamic stability while preserving fast, unguided inference. Future directions: single-objective alignment, smaller preference models.

## Code Availability

Code and pretrained checkpoints: `https://github.com/gersteinlab/elign`.

## Funding

Supported in part by the A.L. Williams Professorship funds.

## 5. Related Work.

**Diffusion Models for Molecules.** For molecular generation, equivariant diffusion have achieved strong geometric fidelity while respecting symmetries (Hoogeboom et al., 2022; Xu et al., 2023; Cornet et al., 2025; Peng et al., 2023; Song

## Impact Statement

This paper presents work whose goal is to advance the field of Machine Learning. There are many potential societal consequences of our work, none which we feel must be specifically highlighted here.

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

## Appendix Table of Contents

# A. Hyperparameters

**Mapping to paper notation.** In each RL iteration, we sample $N_{\mathrm{grp}}$ shared-prefix groups (each group corresponds to a fixed prefix state $z_{T_{\mathrm{prefix}}}$) and generate $K$ rollouts per group. We use $w_F = $ `reward.force_adv_weight` and $w_E = $ `reward.energy_adv_weight`.

*Table 6.* Hyperparameters for RL post-training (QM9)

| Notation | Description | Value |
|---|---|---|
| `train.learning_rate` | Learning rate for policy optimization | $4 \times 10^{-6}$ |
| `train.clip_range` | PPO clipping threshold | $2 \times 10^{-3}$ |
| `train.train_micro_batch_size` | Micro-batch size for PPO updates | 8 |
| `train.gradient_accumulation_steps` | Gradient accumulation steps per PPO update | 1 |
| `train.epoch_per_rollout` | PPO optimization epochs per rollout batch | 1 |
| `train.kl_penalty_weight` | KL regularization weight (vs. reference policy) | 0.08 |
| `train.adv_clip_max` | Max magnitude for clipped advantages | 5 |
| `train.max_grad_norm` | Gradient norm clipping threshold | 1.0 |
| `dataloader.sample_group_size` | #groups per iteration ($N_{\mathrm{grp}}$) | 4 |
| `dataloader.each_prompt_sample` | $K$ rollouts per group | 6 |
| `dataloader.micro_batch_size` | Rollout-generation micro-batch size | 24 |
| `dataloader.epochs` | Number of rollout/update iterations | 200 |
| `model.time_step` | Diffusion timesteps used during rollouts | 1000 |
| `reward.mlff_model` | MLFF model identifier | `uma-m-1p1` (`uma-s-1p1` optional) |
| `reward.shaping.mlff_batch_size` | Batch size for reward evaluation (chunking) | 32 |
| `reward.force_aggregation` | Aggregation for force errors | `rms` (`max` optional) |
| `reward.force_clip_threshold` | Per-atom force magnitude clip before aggregation | 2.0 |
| `reward.force_adv_weight` | Weight for force advantages ($w_{\mathbf{F}}$ in GRPO mixing) | 1.0 |
| `reward.energy_adv_weight` | Weight for energy advantages ($w_E$ in GRPO mixing) | 0.05 |
| `reward.energy_transform_clip` | Clip transformed energy before negation | 5.0 |
| `reward.shaping.gamma` | Discount factor for shaping returns | 1.0 |
| `reward.shaping.scheduler.skip_prefix` | Steps skipped for shaping schedule ($T - T_{\mathrm{prefix}}$) | 700 |
| `train.scheduler.name` | LR scheduler type | `cosine` |
| `train.scheduler.warmup_steps` | Scheduler warmup steps | 60 |
| `train.scheduler.total_steps` | Scheduler total steps | 1500 |
| `train.scheduler.min_lr_ratio` | Final LR ratio after decay | 0.3 |

*Table 7.* Hyperparameters for RL post-training (GEOM-Drugs)

| Notation | Description | Value |
|---|---|---|
| `train.learning_rate` | Learning rate for policy optimization | $6 \times 10^{-7}$ |
| `train.clip_range` | PPO clipping threshold | 0.03 |
| `train.train_micro_batch_size` | Micro-batch size for PPO updates | 16 |
| `train.gradient_accumulation_steps` | Gradient accumulation steps per PPO update | 8 |
| `train.epoch_per_rollout` | PPO optimization epochs per rollout batch | 1 |
| `train.kl_penalty_weight` | KL regularization weight (vs. reference policy) | 0.2 |
| `train.adv_clip_max` | Max magnitude for clipped advantages | 2 |
| `train.max_grad_norm` | Gradient norm clipping threshold | 0.5 |
| `dataloader.sample_group_size` | #groups per iteration ($N_{\mathrm{grp}}$) | 4 |
| `dataloader.each_prompt_sample` | $K$ rollouts per group | 32 |
| `dataloader.micro_batch_size` | Rollout-generation micro-batch size | 32 |
| `model.time_step` | Diffusion timesteps used during rollouts | 1000 |
| `reward.mlff_model` | MLFF model identifier | `uma-m-1p1` (`uma-s-1p1` optional) |
| `reward.shaping.mlff_batch_size` | Batch size for reward evaluation (chunking) | 16 |
| `reward.force_aggregation` | Aggregation for force errors | `rms` (`max` optional) |
| `reward.force_clip_threshold` | Per-atom force magnitude clip before aggregation | 8.0 |
| `reward.force_adv_weight` | Weight for force advantages ($w_{\mathbf{F}}$ in GRPO mixing) | 0.7 |
| `reward.energy_adv_weight` | Weight for energy advantages ($w_E$ in GRPO mixing) | 0.05 |
| `reward.energy_transform_clip` | Clip transformed energy before negation | 10.0 |
| `reward.shaping.gamma` | Discount factor for shaping returns | 1.0 |
| `reward.shaping.scheduler.skip_prefix` | Steps skipped for shaping schedule ($T - T_{\mathrm{prefix}}$) | 600 |
| `train.scheduler.name` | LR scheduler type | `cosine` |
| `train.scheduler.warmup_steps` | Scheduler warmup steps | 60 |
| `train.scheduler.total_steps` | Scheduler total steps | 1500 |
| `train.scheduler.min_lr_ratio` | Final LR ratio after decay | 0.3 |

*Table 8.* Hyperparameters used for the pretrained model on QM9

| Notation | Description | Value |
|---|---|---|
| model | Dynamics model type | egnn_dynamics |
| probabilistic_model | Probabilistic model type | diffusion |
| diffusion_steps | Number of diffusion steps | 1000 |
| diffusion_noise_schedule | Noise schedule used in diffusion | polynomial_2 |
| diffusion_loss_type | Loss function for diffusion training | l2 |
| diffusion_noise_precision | Minimum diffusion noise precision | $10^{-5}$ |
| n_epochs | Total number of training epochs | 3000 |
| batch_size | Batch size used for training | 64 |
| lr | Learning rate for the optimizer | $10^{-4}$ |
| n_layers | Number of EGNN layers | 9 |
| inv_sublayers | Number of invariant sublayers per EGNN layer | 1 |
| nf | Hidden feature dimension | 256 |
| tanh | Use tanh activation in coordinate MLPs | True |
| attention | Use attention mechanisms in EGNN layers | True |
| actnorm | Use ActNorm | True |
| condition_time | Condition on diffusion timestep | True |
| norm_constant | Normalization constant in coordinate updates | 1 |
| sin_embedding | Use sinusoidal time embeddings | False |
| ode_regularization | ODE regularization strength | $10^{-3}$ |
| dataset | Dataset | qm9 |
| filter_n_atoms | Restrict dataset to fixed number of atoms | None |
| remove_h | Remove hydrogens | False |
| dequantization | Dequantization strategy | argmax_variational |
| ema_decay | EMA decay | 0.9999 |
| n_stability_samples | Samples used to compute stability during eval | 1000 |
| normalize_factors | Normalization factors for x/categorical/integer | [1,4,10] |
| include_charges | Include atom charge feature | True |
| normalization_factor | Normalize sum aggregation of EGNN | 1 |
| aggregation_method | Aggregation for the graph network | sum |

*Table 9.* Hyperparameters used for the pretrained model on GEOM-Drugs

| Notation | Description | Value |
|---|---|---|
| model | Dynamics model type | egnn_dynamics |
| probabilistic_model | Probabilistic model type | diffusion |
| diffusion_steps | Number of diffusion steps | 1000 |
| diffusion_noise_schedule | Noise schedule used in diffusion | polynomial_2 |
| diffusion_loss_type | Loss function for diffusion training | l2 |
| diffusion_noise_precision | Minimum diffusion noise precision | $10^{-5}$ |
| n_epochs | Total number of training epochs | 3000 |
| batch_size | Batch size used for training | 64 |
| lr | Learning rate for the optimizer | $10^{-4}$ |
| n_layers | Number of EGNN layers | 4 |
| inv_sublayers | Number of invariant sublayers per EGNN layer | 1 |
| nf | Hidden feature dimension | 256 |
| tanh | Use tanh activation in coordinate MLPs | True |
| attention | Use attention mechanisms in EGNN layers | True |
| actnorm | Use ActNorm | True |
| condition_time | Condition on diffusion timestep | True |
| norm_constant | Normalization constant in coordinate updates | 1 |
| sin_embedding | Use sinusoidal time embeddings | False |
| ode_regularization | ODE regularization strength | $10^{-3}$ |
| dataset | Dataset | geom |
| filter_n_atoms | Restrict dataset to fixed number of atoms | None |
| remove_h | Remove hydrogens | False |
| dequantization | Dequantization strategy | argmax_variational |
| ema_decay | EMA decay | 0.9999 |
| n_stability_samples | Samples used to compute stability during eval | 500 |
| normalize_factors | Normalization factors for x/categorical/integer | [1,4,10] |
| include_charges | Include atom charge feature | False |
| normalization_factor | Normalize sum aggregation of EGNN | 1.0 |
| aggregation_method | Aggregation for the graph network | sum |

# B. Notations

*Table 10.* Summary of symbols and notation used throughout the paper and appendix.

| Symbol | Type | Description |
|---|---|---|
| **Geometric states and molecular structure** | | |
| $\mathbf{x}_0$ | $\mathbb{R}^{N \times 3}$ | Clean atomic coordinates |
| $\mathbf{x}_t$ | $\mathbb{R}^{N \times 3}$ | Noisy conformation at diffusion time $t$ |
| $N$ | $\mathbb{N}$ | Number of atoms |
| $\mathbf{z}_t$ | $\mathbb{R}^{N \times 3} \times \mathbb{R}^{N \times d_h}$ | Diffusion state (coordinates plus hidden features) |
| **Diffusion process and generative model** | | |
| $p_t(\mathbf{z})$ | distribution | Marginal distribution at time $t$ |
| $p_\theta(\mathbf{z}_{t-1} \mid \mathbf{z}_t)$ | distribution | Reverse diffusion transition |
| $s_\theta(\mathbf{z}_t, t)$ | $\mathbb{R}^{N \times 3} \times \mathbb{R}^{N \times d_h}$ | Score function $\nabla_{\mathbf{z}_t} \log p_t(\mathbf{z}_t)$ |
| $s_\theta^{(\mathbf{x})}(\mathbf{z}_t, t)$ | $\mathbb{R}^{N \times 3}$ | Position component of the score |
| $\boldsymbol{\epsilon}$ | $\mathbb{R}^{N \times 3}$ | Standard Gaussian noise |
| $\alpha_t$ | $\mathbb{R}$ | Signal coefficient |
| $\sigma_t$ | $\mathbb{R}_{>0}$ | Noise scale |
| **Trajectories and expectations** | | |
| $\tau$ | sequence | Reverse diffusion trajectory $(\mathbf{z}_T, \ldots, \mathbf{z}_0)$ |
| $\mathbb{E}$ | operator | Expectation over samples or trajectories |
| **Energy, forces, and physical models** | | |
| $E(z)$ | $\mathbb{R}$ | Potential energy of a clean structure $z$ |
| $\mathbf{F}(z)$ | $\mathbb{R}^{N \times 3}$ | Force field $-\nabla_{\mathbf{x}} E(z)$ |
| $E_\star(z)$ | $\mathbb{R}$ | Target energy (gold standard reference) |
| $E_\phi(z)$ | $\mathbb{R}$ | MLFF energy approximation |
| $\mathbf{F}_\star(z)$ | $\mathbb{R}^{N \times 3}$ | Target forces $-\nabla_{\mathbf{x}} E_\star(z)$ |
| $\mathbf{F}_\phi(z)$ | $\mathbb{R}^{N \times 3}$ | MLFF forces $-\nabla_{\mathbf{x}} E_\phi(z)$ |
| $\|\mathbf{F}(z)\|_{\mathrm{rms}}$ | $\mathbb{R}_{\geq 0}$ | RMS force magnitude, $\|\mathbf{F}(z)\|_{\mathrm{rms}} := \frac{1}{\sqrt{N}} \|\mathbf{F}(z)\|_F$ |
| **Reinforcement learning and GRPO** | | |
| $\pi_\theta$ | policy | Reverse diffusion policy |
| $r_t^E$ | $\mathbb{R}$ | Energy based reward |
| $r_t^F$ | $\mathbb{R}$ | RMS force based stability reward |
| $r_t$ | $\mathbb{R}$ | Total reward |
| $\hat{A}_t$ | $\mathbb{R}$ | Advantage estimate |
| $\mathcal{L}_{\mathrm{GRPO}}$ | objective | Group Relative Policy Optimization loss |
| $\mathcal{L}_{\mathrm{GRPO\text{-}disent}}$ | objective | Disentangled GRPO loss (energy and force) |
| $\varepsilon$ | $\mathbb{R}_{>0}$ | PPO clipping threshold (clip range) |
| **Measure theoretic and variational notation (appendix proofs)** | | |
| $\mathcal{P}(\mathcal{X})$ | space | Probability measures on $\mathcal{X}$ |
| $\mu$ | $\mathcal{P}(\mathcal{X})$ | Pretrained terminal law, $\mu := \rho_{\theta_{\mathrm{pre}}}$ |
| $\rho$ | $\mathcal{P}(\mathcal{X})$ | Candidate terminal law |
| $\rho \ll \mu$ | relation | Absolute continuity of $\rho$ with respect to $\mu$ |
| $\mathrm{KL}(\rho \| \mu)$ | $\mathbb{R}_{\geq 0}$ | Kullback Leibler divergence |
| $\mathcal{J}(\rho)$ | functional | Variational objective over terminal laws |
| $w_E$ | $\mathbb{R}_{>0}$ | Energy weight in advantage calculation |
| $w_{\mathrm{KL}}$ | $\mathbb{R}_{>0}$ | KL weight in $\mathcal{J}$ |
| $\beta_{\mathrm{eff}}$ | $\mathbb{R}_{>0}$ | Effective inverse temperature, $\beta_{\mathrm{eff}} := 1/w_{\mathrm{KL}}$ |
| $\rho^\star$ | $\mathcal{P}(\mathcal{X})$ | Maximizer of $\mathcal{J}$ |
| $Z_\phi$ | $\mathbb{R}_{>0}$ | Partition function for $\rho^\star$ under $E_\phi$ |
| $Z_\star$ | $\mathbb{R}_{>0}$ | Partition function for the tilt under $E_\star$ |
| $f$ | density | Radon Nikodym derivative $f := d\rho/d\mu$ |
| $\lambda$ | $\mathbb{R}$ | Lagrange multiplier for normalization of $f$ |
| $\|\cdot\|_{\mathrm{TV}}$ | $\mathbb{R}_{\geq 0}$ | Total variation distance |
| $\delta$ | $\mathbb{R}_{\geq 0}$ | Uniform energy error bound, $\sup_{z \in \mathcal{X}} |E_\phi(z) - E_\star(z)| \leq \delta$ |

# C. Related Works

**Diffusion Models for Molecules.** E(3)-equivariant diffusion models have quickly become a leading paradigm for 3D molecular generation, as they inherently enforce rotational and translational symmetries and produce geometries with high fidelity. (Hoogeboom et al., 2022) introduced an SE(3)-equivariant diffusion process that generates molecules as sets of atoms with remarkable realism. Building on this foundation, (Xu et al., 2023) developed GeoLDM, a latent 3D diffusion approach that operates in a learned point-cloud latent space composed of both invariant scalars and equivariant tensors, improving sample efficiency and controllable generation of molecular geometries. More recently, (Cornet et al., 2025) proposed an Equivariant Neural Diffusion (END) model that also learns the forward-noising process, achieving state-of-the-art results on standard molecular generation benchmarks. Beyond unconditional generation, many methods aim to guide or bias diffusion models toward molecules with desired properties or constraints. One strategy is to inject domain knowledge via physically-informed priors. (Wu et al., 2022) introduces informative prior bridges that steer the diffusion trajectory toward low-energy conformations by constructing a biased intermediate distribution for the sampler. (Zhou et al., 2025) formulates 3D molecule generation as a Markov decision process and uses proximal policy optimization to fine-tune a pre-trained equivariant diffusion model's sampling policy by providing a reward based on physics evaluations. An alternative formalism is to cast guided generation as a stochastic optimal control problem. (Havens et al., 2025) introduces Adjoint Sampling, which optimizes a matching objective for the diffusion's probability flow by learning an optimal diffusion drift that samples from an unnormalized target density, such as a Boltzmann distribution. (Liu et al., 2025a) generalizes this idea by relaxing previous assumptions on the prior, solving a Schrödinger Bridge between a simple base distribution and the desired Boltzmann-like distribution.

**Machine-learning Force-Field.** Machine learning force fields (MLFFs) use geometric deep learning to approximate potential energy surfaces and interatomic forces with high accuracy at far lower cost than quantum chemistry. Early MLFFs relied on invariant message passing networks that predict an energy scalar invariant to rotations and translations, with forces obtained via differentiation. SchNet established this paradigm using continuous filter convolutions over interatomic distances (Schütt et al., 2018). More generally, graph network formalisms demonstrated that message passing can unify molecular and crystalline property prediction under a single framework (Chen et al., 2019). A major line of progress then focused on injecting richer geometric structure into invariant architectures. DimeNet introduced directional message passing through angular features and specialized basis expansions (Gasteiger et al., 2020), and GemNet improved higher order interaction modeling to better capture many-body effects relevant for energies and forces (Gasteiger et al., 2021). SphereNet further leveraged a spherical coordinate parameterization for local neighborhoods to encode distances and angles in a unified way (Liu et al., 2022), while ComENet emphasized complete yet efficient geometric message passing to improve accuracy without sacrificing scaling (Wang et al., 2022). Recent analyses underscore that architecture and training data choices critically determine transfer across chemical domains and system sizes, motivating careful study of inductive biases and data coverage (Qu & Krishnapriyan, 2024). In parallel, E(3) equivariant MLFFs enforce symmetry directly in intermediate representations, enabling vector and tensor features that transform correctly under rotations and translations. EGNN provides a lightweight recipe for equivariant message passing with coordinate updates (Satorras et al., 2021), and complete local frame constructions strengthen SE(3) equivariance and stability by anchoring messages in local geometric frames (Du et al., 2022). Tensor-based designs such as TensorNet offer an efficient Cartesian tensor alternative to spherical harmonic pipelines while preserving equivariant structure (Simeon & De Fabritiis, 2024), and newer architectures focus explicitly on practical efficiency, such as GotenNet (Aykent & Xia, 2025). Equivariant graph networks have also been used as surrogates for more demanding electronic structure components, improving the scalability of learned approximations to Kohn-Sham DFT quantities while maintaining symmetry constraints (Li et al., 2024). A particularly influential family of MLFFs builds on SO(3) representation theory with spherical harmonics. SE(3) Transformer introduced equivariant attention with irreducible representations (Fuchs et al., 2020), and Equiformer extended this transformer style approach with strong results on quantum chemistry benchmarks (Liao & Smidt, 2023). NequIP demonstrated striking data efficiency for learning interatomic potentials with spherical tensor message passing (Batzner et al., 2022), while MACE advanced higher order equivariant interactions for accurate molecular dynamics potentials (Batatia et al., 2022). Scalability has been further improved with locally deployed equivariant potentials such as Allegro (Musaelian et al., 2023), extensions that model both long range and short range interactions (Li et al., 2023), and methods that reduce the cost of equivariant operations (Passaro & Zitnick, 2023; Li et al., 2025). Finally, recent work targets improved physical consistency and broader transfer, including learning objectives tuned for stable molecular dynamics and phonon properties (Fu et al., 2025) and universal pretrained atomistic models that aim to generalize across elements and compounds (Wood et al., 2025).

**Reinforcement Learning for Diffusion Guidance.** Integrating RL with diffusion models is a recent trend aimed at steering generative models toward complex objectives that are hard to encode in the training likelihood. One representative approach is Denoising Diffusion Policy Optimization (DDPO) (**?**). DDPO reframes the multi-step denoising process as a Markov decision process, where each diffusion step is an action, and a reward is obtained at the final sample. This allows applying policy gradient algorithms to fine-tune a pretrained diffusion model for higher rewards. (Xue et al., 2025) adapts GRPO to diffusion and continuous flow models with *DanceGRPO*, achieving robust RL-based fine-tuning on large-scale text-to-image and text-to-video generation tasks. (Liu et al., 2025b) introduces *Flow-GRPO*, an online RL algorithm that trains flow matching models using reward signals by aligning generated trajectories with desired outcomes through policy gradient updates. (Shen et al., 2024) proposes a chemistry-inspired guided diffusion that uses external force-field evaluations as a form of reward to bias the diffusion sampler toward low-energy structures, without requiring those evaluations to be differentiable. (Zhou et al., 2025) applies Reinforcement Learning with Physical Feedback (RLPF) to an equivariant diffusion model, significantly improving stability of generated conformations. Crucially, both works demonstrate that RL can incorporate domain-specific criteria (like force-field energies) that lie outside the scope of the original generative model's training. Another perspective comes from control theory: *Adjoint methods* avoid explicit RL by solving optimal control formulations for diffusion. (Havens et al., 2025) derives an adjoint formulation to directly match a diffusion sampler to an optimal policy in a single backward pass, while (Liu et al., 2025a) develops an adjoint Schrödinger bridge that computes an optimal transport between the prior and target distributions. These methods achieve goals similar to RL-guided diffusion by biasing generation toward certain distributions or rewards, but do so by analytically aligning sampling trajectories, thereby improving efficiency and scalability.

## D. Proofs for Section 3.2

**Theorem 3** (Energy-aligned terminal distribution (restated)). *Assume $w_{\mathrm{KL}} > 0$ and consider the functional over $\rho \in \mathcal{P}(\mathcal{X})$,*

$$\mathcal{J}(\rho) := \mathbb{E}_{z \sim \rho}[-E_\phi(z)] - w_{\mathrm{KL}} \, \mathrm{KL}(\rho \| \mu), \qquad \textit{with the convention } \mathcal{J}(\rho) = -\infty \textit{ if } \rho \not\ll \mu.$$

*Assume the supremum of $\mathcal{J}$ over $\mathcal{P}(\mathcal{X})$ is attained. Then the maximizer is unique and equals*

$$\rho^\star(dz) = \frac{1}{Z_\phi} \exp(-\beta_{\mathrm{eff}} E_\phi(z)) \, \mu(dz), \qquad \beta_{\mathrm{eff}} := \frac{1}{w_{\mathrm{KL}}}, \qquad Z_\phi := \int_{\mathcal{X}} \exp(-\beta_{\mathrm{eff}} E_\phi(z)) \, \mu(dz).$$

*Proof.* Let $\rho \in \mathcal{P}(\mathcal{X})$. If $\rho \not\ll \mu$, then $\mathrm{KL}(\rho \| \mu) = +\infty$ and $\mathcal{J}(\rho) = -\infty$, so we restrict to $\rho \ll \mu$. Write the Radon–Nikodym derivative $f := d\rho/d\mu$. Then $f \geq 0$ $\mu$-a.e. and $\int f \, d\mu = 1$. Moreover,

$$\mathrm{KL}(\rho \| \mu) = \int_{\mathcal{X}} f(z) \log f(z) \, \mu(dz), \qquad \mathbb{E}_{z \sim \rho}[E_\phi(z)] = \int_{\mathcal{X}} E_\phi(z) \, f(z) \, \mu(dz).$$

Thus maximizing $\mathcal{J}(\rho)$ over $\rho \ll \mu$ is equivalent to maximizing over densities $f$:

$$\mathcal{J}(f) = -\int_{\mathcal{X}} E_\phi(z) \, f(z) \, \mu(dz) - w_{\mathrm{KL}} \int_{\mathcal{X}} f(z) \log f(z) \, \mu(dz), \quad \text{s.t.} \int_{\mathcal{X}} f(z) \, \mu(dz) = 1, \; f \geq 0.$$

Introduce a Lagrange multiplier $\lambda \in \mathbb{R}$ for the normalization constraint and define

$$\mathcal{L}(f, \lambda) = -\int E_\phi(z) f(z) \, \mu(dz) - w_{\mathrm{KL}} \int f(z) \log f(z) \, \mu(dz) + \lambda \left( \int f(z) \, \mu(dz) - 1 \right).$$

Let $h$ be any bounded perturbation with $\int h \, d\mu = 0$ and consider $f_\varepsilon = f + \varepsilon h$. A first-order stationarity condition at an optimum $f^\star$ implies $\frac{d}{d\varepsilon}\mathcal{L}(f^\star + \varepsilon h, \lambda)|_{\varepsilon=0} = 0$ for all such $h$. Using $\frac{d}{d\varepsilon}[(f^\star + \varepsilon h) \log(f^\star + \varepsilon h)]|_{\varepsilon=0} = h(1 + \log f^\star)$, we obtain

$$0 = \int_{\mathcal{X}} h(z) \Big( -E_\phi(z) - w_{\mathrm{KL}}(1 + \log f^\star(z)) + \lambda \Big) \mu(dz).$$

Since this holds for all $h$ with zero $\mu$-mean, the bracketed term must be $\mu$-a.e. constant, i.e.

$$-E_\phi(z) - w_{\mathrm{KL}}(1 + \log f^\star(z)) + \lambda = 0 \quad \text{for } \mu\text{-a.e. } z.$$

Rearranging gives

$$\log f^\star(z) = -\frac{1}{w_{\mathrm{KL}}} E_\phi(z) + c = -\beta_{\mathrm{eff}} E_\phi(z) + c$$

for some constant $c \in \mathbb{R}$, hence

$$f^\star(z) = \exp(c) \exp\big( -\beta_{\mathrm{eff}} E_\phi(z)\big).$$

Imposing $\int f^\star \, d\mu = 1$ yields $\exp(c) = 1/Z_\phi$ with

$$Z_\phi = \int_{\mathcal{X}} \exp\big( -\beta_{\mathrm{eff}} E_\phi(z)\big) \, \mu(dz),$$

so $\rho^\star(dz) = f^\star(z)\mu(dz)$ as claimed.

Uniqueness follows because $f \mapsto \int f \log f \, d\mu$ is strictly convex on $\{f \geq 0 : \int f \, d\mu = 1\}$, hence $\mathcal{J}(f)$ is strictly concave when $w_{\mathrm{KL}} > 0$. $\qquad\square$

**MLFF approximation and distribution error.** Let $E_\star : \mathcal{X} \to \mathbb{R}$ be a target energy and $E_\phi$ its MLFF approximation. Define the two tilted laws (with the same reference $\mu$):

$$\rho_\star^\star(dz) = \frac{1}{Z_\star} \exp\big( -\beta_{\mathrm{eff}} E_\star(z)\big) \, \mu(dz), \qquad \rho_\phi^\star(dz) = \frac{1}{Z_\phi} \exp\big( -\beta_{\mathrm{eff}} E_\phi(z)\big) \, \mu(dz),$$

where

$$Z_\star := \int_{\mathcal{X}} \exp\big( -\beta_{\mathrm{eff}} E_\star(z)\big) \, \mu(dz), \qquad Z_\phi := \int_{\mathcal{X}} \exp\big( -\beta_{\mathrm{eff}} E_\phi(z)\big) \, \mu(dz).$$

**Theorem 4** (Energy error implies distribution error (restated)). *Assume $\sup_{z \in \mathcal{X}} |E_\phi(z) - E_\star(z)| \leq \delta$ for some $\delta \geq 0$. Then*

$$\|\rho_\phi^\star - \rho_\star^\star\|_{\mathrm{TV}} \leq \tanh(\beta_{\mathrm{eff}}\delta).$$

**Lemma 1** (Likelihood-ratio bound implies TV bound). *Let $P, Q$ be probability measures with $P \ll Q$ and suppose there exists $\varepsilon \geq 0$ such that*

$$e^{-\varepsilon} \leq \frac{dP}{dQ} \leq e^{\varepsilon} \quad Q\text{-a.e.}$$

*Then $\|P - Q\|_{\mathrm{TV}} \leq \tanh(\varepsilon/2)$.*

*Proof.* Let $r := dP/dQ$, so $r \in [e^{-\varepsilon}, e^{\varepsilon}]$ $Q$-a.e. and $\mathbb{E}_Q[r] = 1$. Let $A := \{r \geq 1\}$. Then

$$\|P - Q\|_{\mathrm{TV}} = \frac{1}{2} \int |dP - dQ| = \frac{1}{2} \int |r - 1| \, dQ = \int_A (r - 1) \, dQ,$$

where the last equality uses $\int_A (r - 1) \, dQ = \int_{A^c} (1 - r) \, dQ$ from $\mathbb{E}_Q[r] = 1$.

Write $q := Q(A)$ and the conditional means $m_+ := \mathbb{E}_Q[r \mid A] \in [1, e^{\varepsilon}]$, $m_- := \mathbb{E}_Q[r \mid A^c] \in [e^{-\varepsilon}, 1]$. The constraint $\mathbb{E}_Q[r] = 1$ becomes $qm_+ + (1 - q)m_- = 1$, and the TV becomes $q(m_+ - 1)$. The maximum TV under the bounds is achieved at the extreme values $m_+ = e^{\varepsilon}$ and $m_- = e^{-\varepsilon}$. Solving

$$qe^{\varepsilon} + (1 - q)e^{-\varepsilon} = 1 \quad \implies \quad q = \frac{1}{e^{\varepsilon} + 1},$$

we get

$$\|P - Q\|_{\mathrm{TV}} \leq q(e^{\varepsilon} - 1) = \frac{e^{\varepsilon} - 1}{e^{\varepsilon} + 1} = \tanh(\varepsilon/2).$$

$\qquad\square$

*Proof of Theorem 4.* Let $\beta := \beta_{\mathrm{eff}}$. Define $w_\star(z) := \exp(-\beta E_\star(z))$ and $w_\phi(z) := \exp(-\beta E_\phi(z))$. By $|E_\phi(z) - E_\star(z)| \leq \delta$, for all $z \in \mathcal{X}$,

$$e^{-\beta\delta} w_\star(z) \leq w_\phi(z) \leq e^{\beta\delta} w_\star(z).$$

Integrating w.r.t. $\mu$ yields the partition-function sandwich:

$$e^{-\beta\delta} Z_\star \leq Z_\phi \leq e^{\beta\delta} Z_\star.$$

Therefore, for $\rho_\star^\star$-a.e. $z$,

$$\frac{d\rho_\phi^\star}{d\rho_\star^\star}(z) = \frac{Z_\star}{Z_\phi} \cdot \frac{w_\phi(z)}{w_\star(z)} \in [e^{-\beta\delta}, e^{\beta\delta}] \cdot [e^{-\beta\delta}, e^{\beta\delta}] = [e^{-2\beta\delta}, e^{2\beta\delta}].$$

Applying Lemma 1 with $P = \rho_\phi^\star$, $Q = \rho_\star^\star$ and $\varepsilon = 2\beta\delta$ gives

$$\|\rho_\phi^\star - \rho_\star^\star\|_{\mathrm{TV}} \leq \tanh((2\beta\delta)/2) = \tanh(\beta\delta) = \tanh(\beta_{\mathrm{eff}}\delta).$$

$\square$

# E. Supplementary Experiments

## E.1. Qualitative analysis.

We provide qualitative examples on GEOM-Drugs to illustrate how post-training affects both 3D geometry and validity.

**Figure 7: Geometry overlays.**    We overlay generated conformers (colored) with the corresponding reference structure (gray). Compared to the baseline diffusion model, the post-trained sampler more consistently produces coherent, well-connected geometries that better match the reference conformer.

**Figure 8: Additional energy/force annotations.**    Beyond the conformer-quality teaser shown in the main text (Fig. 6), we provide further representative examples with MLFF-predicted energy and RMS force norms. Post-training typically shifts samples toward lower energies and smaller residual forces, consistent with configurations closer to local equilibrium.

**Figure 9: Matched-noise denoising trajectories.**    Using the same initial noise, we compare reverse diffusion trajectories and show that post-training corrects failure modes that persist throughout denoising in the baseline (e.g., distorted rings or strained geometries), yielding a chemically valid final structure at $t = 0$.

**Figure 10: Invalid-to-valid corrections.**    We highlight cases where the baseline produces chemically invalid structures, while the post-trained model generates valid conformers for the same molecular graphs.

Together, these figures show that alignment improves terminal quality and also stabilizes intermediate geometries along the denoising trajectory.

## E.2. Property controllability and predictor-based conditional accuracy.

We split the QM9 training set into two disjoint halves with 50k samples each. We train the property prediction network $\omega$ on the first half, and train conditional generative models on the second half. We sample target property values $\mathbf{c}$ from the held-out QM9 property distribution, generate conditional samples $\mathbf{z}_0 \sim \rho_\theta(\cdot \mid \mathbf{c})$, and compute predicted properties $\hat{\mathbf{c}}(\mathbf{z}_0) := \omega(\mathbf{z}_0)$. We then report the MAE to the conditioning target, i.e., $\mathbb{E} \|\hat{\mathbf{c}}(\mathbf{z}_0) - \mathbf{c}\|_1$ (not to an external QC oracle).

*Property alignment reward.* For *conditional generation* with a target property vector $\mathbf{y}$, we add an auxiliary *property alignment reward* at the terminal state:

$$r_0^{(P)}(\mathbf{z}_0; \mathbf{y}) := -\|\omega(\mathbf{z}_0) - \mathbf{y}\|_1. \tag{9}$$

The total terminal reward combines $r_0^{(P)}$ with the energy/force rewards used by Elign. The property reward is treated as an additional disentangled channel (analogous to energy and force) and is *group-normalized separately* before being weighted and added to the total advantage.

*Results.* As shown in Table 11, Elign substantially reduces MAE compared to unguided diffusion baselines, narrowing the gap toward the QM9 oracle. Random samples define an upper bound on MAE, while QM9 ground-truth molecules provide a lower bound. Lower MAE indicates better conditional controllability/accuracy with respect to the learned predictor $\omega$ (we do not evaluate against an external quantum-chemistry oracle here). We report results for isotropic polarizability $\alpha$,

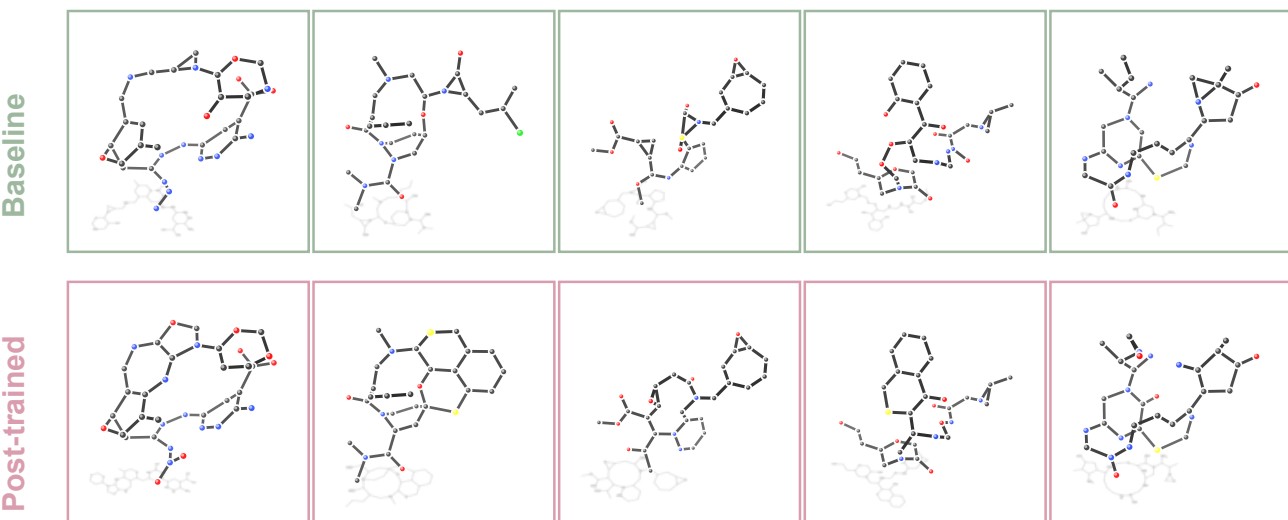

*Figure 7.* Geometry overlays on GEOM-Drugs. Top row: baseline diffusion model (green). Bottom row: post-trained model (pink). In each panel, the generated conformer (dark) is overlaid on the reference structure (light gray). Post-training improves connectivity and overall geometric fidelity to the reference conformer.

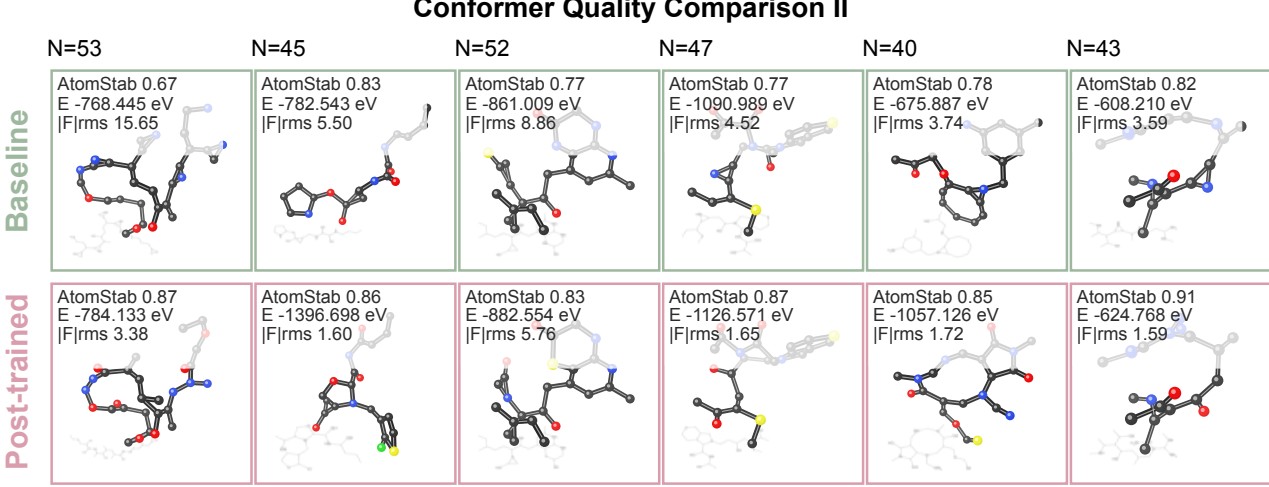

*Figure 8.* Additional conformer-quality examples with MLFF metrics. Top row: baseline model. Bottom row: post-trained model. Post-training reduces MLFF-predicted energy ($E$) and RMS force norms across examples. Generated conformers (dark) are overlaid on reference structures (light gray).

HOMO–LUMO gap $\Delta\varepsilon$, and LUMO energy $\varepsilon_{\text{LUMO}}$. Notably, Elign improves all properties simultaneously, indicating that alignment with MLFF-derived energy and force feedback yields samples that are not only structurally stable but also chemically predictive.

### E.3. More Ablation Studies

**GRPO component ablations.** Table 12 ablates key ingredients of our GRPO procedure. **Disentangled normalization** matters because the force channel can otherwise dominate the combined advantage (due to scale), which biases learning toward mechanically "quiet" but less diverse solutions and hurts the validity–uniqueness trade-off (V×U). **Shared-prefix grouping** matters because, without conditioning comparisons on a common partially denoised prefix, the group-relative baseline has higher variance and the resulting noisy advantages degrade learning. Finally, **dense PBRS shaping** matters because without it the reward becomes effectively sparse along the diffusion trajectory, slowing credit assignment and weakening optimization over long horizons.

**Matched-noise diffusion trajectory (N=60)**

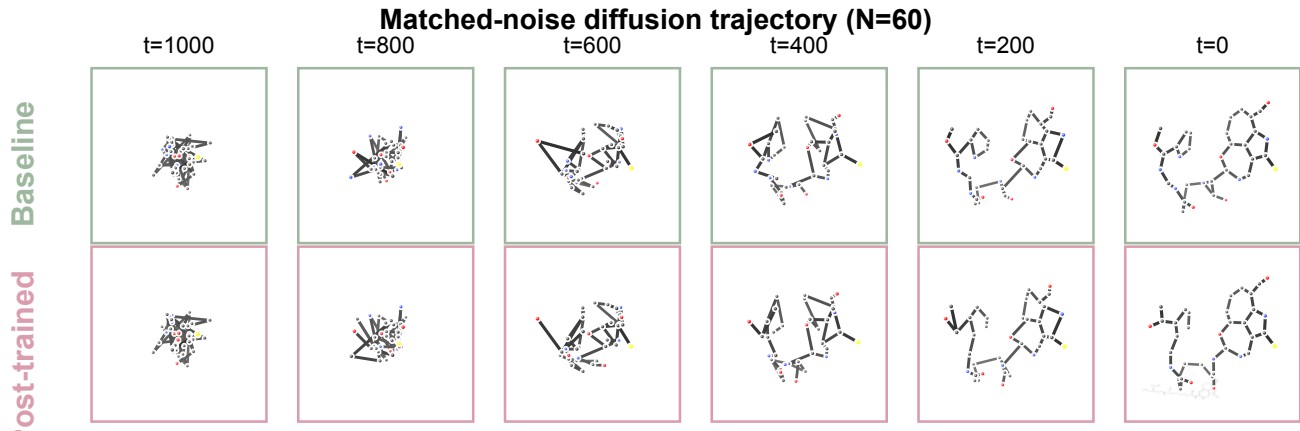

*Figure 9.* Matched-noise reverse diffusion trajectories on GEOM-Drugs (same initial noise; $N = 60$ atoms, seed=89). Top row: baseline (green). Bottom row: post-trained (pink). Snapshots are shown at timesteps $t \in \{1000, 800, 600, 400, 200, 0\}$. The baseline trajectory retains distorted rings and strained geometries and ends in an invalid structure at $t = 0$, whereas the post-trained model corrects these failures and produces a valid conformer.

**Invalid → valid examples**

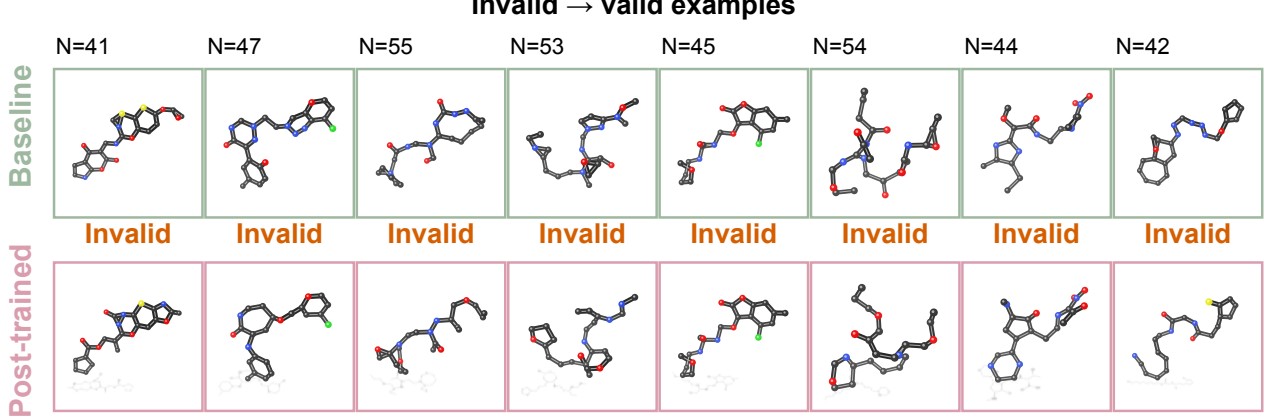

*Figure 10.* **Invalid-to-valid corrections on GEOM-Drugs.** Each column shows the same molecule ($N$ atoms) generated by the baseline model (top row, green border) and the post-trained model (bottom row, pink border). The baseline produces chemically invalid structures, while post-training yields valid conformers for the same molecular graphs. Atom colors: carbon (gray), nitrogen (blue), oxygen (red), sulfur (yellow), chlorine (green); hydrogens are small gray spheres.

*Table 11.* Mean Absolute Error (MAE) for molecular property prediction. Lower values indicate better controllable generation. Properties are predicted by a pretrained E(3)-equivariant EGNN regressor $\omega$ on molecular samples generated by each method. QM9 and Random serve as lower and upper bounds, respectively.

| Property | $\alpha$ | $\Delta\varepsilon$ | $\varepsilon_{\text{LUMO}}$ |
| Units | Bohr$^3$ | meV | meV |
| --- | --- | --- | --- |
| QM9 | 0.10 | 64 | 36 |
| Random | 9.01 | 1470 | 1457 |
| $N_{\text{atoms}}$ | 3.86 | 866 | 813 |
| EDM (Hoogeboom et al., 2022) | 2.76 | 655 | 584 |
| GeoLDM (Xu et al., 2023) | 2.37 | 587 | 522 |
| GeoBFN (Song et al., 2024) | 2.34 | 577 | 516 |
| Elign (w/ Additional Property Alignment Reward) | **2.32** | **564** | **512** |

*Table 12.* GRPO component ablations.

| Setting | Disentangle? | Shared prefix? | PBRS? | A↑ | M↑ | V×U↑ |
|---|---|---|---|---|---|---|
| Full GRPO (Elign) | ✓ | ✓ | ✓ | 99.33 | 93.70 | 95.31 |
| No disentanglement | × | ✓ | ✓ | 99.12 | 92.60 | 93.80 |
| No shared prefix grouping | ✓ | × | ✓ | 98.85 | 83.40 | 91.20 |
| Naive per-step energy (no PBRS) | ✓ | ✓ | × | 99.20 | 92.50 | 94.60 |

## F. Supplementary Theory: PBRS as an Alchemical Force

While potential-based reward shaping (PBRS) is commonly introduced as a credit-assignment heuristic, in our setting it admits a concrete physical interpretation. We show that PBRS induces an exponential tilting of the one-step reverse diffusion kernel under a KL-regularized local control view, and that in the small-step limit this tilt manifests as an additional drift term proportional to the gradient of the shaping potential. When the potential is chosen as the negative MLFF energy evaluated on the predicted clean geometry, this drift corresponds to a force-like correction acting on the reverse dynamics. Crucially, because the diffusion state includes both coordinates and relaxed feature channels, the induced term decomposes into a physical force on atomic positions and an auxiliary "alchemical" force on non-geometric variables. This result formalizes PBRS as a principled approximation to energy-guided diffusion, clarifying how dense shaping rewards compile physical guidance into the learned reverse policy at training time, rather than injecting oracle forces during inference

This interpretation assumes a continuous relaxation (and differentiable $E_\phi$); the RL algorithm itself does not require reward differentiability.

**Theorem 5** (PBRS induces an approximate (alchemical) force in the reverse drift). *Fix a reverse step $t$ and denote the shaping potential by $\Psi_t(z) := -E_\phi(\hat{z}_{0|t}(z))$. Let the base one-step reverse kernel be Gaussian,*

$$q_t^{\Delta t}(z_{t-\Delta t} \mid z_t) = \mathcal{N}(z_{t-\Delta t};\ z_t + b_t(z_t)\Delta t,\ a_t\Delta t),$$

*(e.g. Euler–Maruyama for a reverse SDE; $a_t$ may include $\mathbf{P}_{\mathrm{CoM}}$). Consider the per-step KL-regularized local update with PBRS reward $r_t^{\mathrm{shape}} = \gamma\Psi_{t-\Delta t}(z_{t-\Delta t}) - \Psi_t(z_t)$ (Eq. (3)):*

$$\pi_t^{\star,\Delta t}(\cdot \mid z_t) \in \arg\max_{\pi(\cdot\mid z_t)} \left\{ \mathbb{E}_\pi[r_t^{\mathrm{shape}}] - w_{\mathrm{KL}}\mathrm{KL}(\pi(\cdot \mid z_t)\|q_t^{\Delta t}(\cdot \mid z_t)) \right\}.$$

*Assume $\Psi_t$ is $C^1$ in $z$ with locally Lipschitz $\nabla\Psi_t$. Then (i) the optimizer is the exponential tilt*

$$\pi_t^{\star,\Delta t}(z_{t-\Delta t} \mid z_t) \propto q_t^{\Delta t}(z_{t-\Delta t} \mid z_t)\exp\left(\frac{\gamma}{w_{\mathrm{KL}}}\Psi_{t-\Delta t}(z_{t-\Delta t})\right),$$

*and (ii) its conditional mean admits the small-step expansion*

$$\mathbb{E}_{\pi_t^{\star,\Delta t}}[z_{t-\Delta t} \mid z_t] = z_t + \left(b_t(z_t) + \frac{\gamma}{w_{\mathrm{KL}}}\,a_t\nabla\Psi_t(z_t)\right)\Delta t + o(\Delta t).$$

*Equivalently, in the $\Delta t \to 0$ limit the induced controlled reverse drift is*

$$\tilde{b}_t(z) = b_t(z) + \frac{\gamma}{w_{\mathrm{KL}}}\,a_t\nabla\Psi_t(z) = b_t(z) - \frac{\gamma}{w_{\mathrm{KL}}}\,a_t\nabla_z E_\phi(\hat{z}_{0|t}(z)).$$

*Writing $z = [x, h]$, the $x$-block recovers a force-like term $(-\nabla_x E_\phi)$ while the $h$-block yields an "alchemical" preference direction $(-\nabla_h E_\phi)$ on relaxed types/features.*

*Proof.* Fix $z_t$. The term $-\Psi_t(z_t)$ is constant in $z_{t-\Delta t}$, so the objective is

$$\max_\pi \left\{ \gamma\,\mathbb{E}_\pi[\Psi_{t-\Delta t}(z_{t-\Delta t})] - w_{\mathrm{KL}}\mathrm{KL}(\pi\|q_t^{\Delta t}) \right\}.$$

A one-line variational calculation (Lagrange multiplier for $\int\pi = 1$) gives the unique maximizer

$$\pi_t^{\star,\Delta t}(u \mid z_t) \propto q_t^{\Delta t}(u \mid z_t)\exp(\eta\Psi_{t-\Delta t}(u)), \qquad \eta := \gamma/w_{\mathrm{KL}}.$$

Now use Gaussian integration by parts (Stein's identity): if $U \sim \mathcal{N}(\mu, \Sigma)$ and $g$ is smooth, $\mathbb{E}[(U - \mu)g(U)] = \Sigma \, \mathbb{E}[\nabla g(U)]$. Apply this with $U \sim q_t^{\Delta t}(\cdot \mid \boldsymbol{z}_t)$ and $g(u) = \exp(\eta \Psi_{t-\Delta t}(u))$ to obtain

$$\mathbb{E}_{\pi_t^{\star}, \Delta t}[U \mid \boldsymbol{z}_t] = \mu + \Sigma \, \eta \, \mathbb{E}_{\pi_t^{\star}, \Delta t}[\nabla \Psi_{t-\Delta t}(U) \mid \boldsymbol{z}_t],$$

where $\mu = \boldsymbol{z}_t + b_t(\boldsymbol{z}_t)\Delta t$ and $\Sigma = a_t \Delta t$. Since $U = \boldsymbol{z}_t + O_{\mathbb{P}}(\sqrt{\Delta t})$ under the Gaussian kernel and $\nabla \Psi$ is locally Lipschitz,

$$\mathbb{E}_{\pi_t^{\star}, \Delta t}[\nabla \Psi_{t-\Delta t}(U) \mid \boldsymbol{z}_t] = \nabla \Psi_t(\boldsymbol{z}_t) + o(1).$$

Substituting and collecting terms yields

$$\mathbb{E}_{\pi_t^{\star}, \Delta t}[\boldsymbol{z}_{t-\Delta t} \mid \boldsymbol{z}_t] = \boldsymbol{z}_t + (b_t(\boldsymbol{z}_t) + \eta \, a_t \nabla \Psi_t(\boldsymbol{z}_t))\Delta t + o(\Delta t),$$

and replacing $\nabla \Psi_t = -\nabla E_\phi(\hat{\boldsymbol{z}}_{0|t})$ gives the stated "force" form. $\qquad \square$

**From theory to measurement.** Theorem 5 provides a local-control view in which energy PBRS induces a force-like correction to the reverse drift. We next empirically probe this effect by measuring the score change induced by post-training and comparing its position component to the MLFF force direction.

### F.1. Alchemical force analysis

To understand how post-training compiles physical preferences into the sampler, we measure an *alchemical force* induced by alignment and compare it to the actual force field. Write $\boldsymbol{z} = [\mathbf{x}, \mathbf{h}]$ for positions and atom-type features. Let $s_{\theta_{\text{pre}}}^{(\mathbf{x})}(\boldsymbol{z}_t, t)$ and $s_\theta^{(\mathbf{x})}(\boldsymbol{z}_t, t)$ denote the *position* components of the pretrained and post-trained score networks, respectively. We define the alchemical force proxy (positions only) as

$$\mathbf{F}_{\text{alc}}(\boldsymbol{z}_t, t) := s_\theta^{(\mathbf{x})}(\boldsymbol{z}_t, t) - s_{\theta_{\text{pre}}}^{(\mathbf{x})}(\boldsymbol{z}_t, t), \tag{10}$$

so that $\mathbf{F}_{\text{alc}} \in \mathbb{R}^{N \times 3}$ is comparable to a physical force field. Given a force oracle (e.g., the MLFF) $\mathbf{F}_\phi(\hat{\boldsymbol{z}}_{0|t})$, we quantify agreement via cosine similarity

$$\cos(\mathbf{F}_{\text{alc}}, \mathbf{F}_\phi) := \frac{\langle \mathbf{F}_{\text{alc}}(\boldsymbol{z}_t, t), \, \mathbf{F}_\phi(\hat{\boldsymbol{z}}_{0|t}) \rangle}{\|\mathbf{F}_{\text{alc}}(\boldsymbol{z}_t, t)\| \, \|\mathbf{F}_\phi(\hat{\boldsymbol{z}}_{0|t})\|}. \tag{11}$$

Here $\hat{\boldsymbol{z}}_{0|t}$ is the predicted clean geometry from the diffusion posterior mean. Empirically, we find that $\mathbf{F}_{\text{alc}}$ is strongly aligned with $\mathbf{F}_\phi$. Figure 11 plots a histogram of $|\cos(\mathbf{F}_{\text{alc}}, \mathbf{F}_\phi)|$ computed over sampled diffusion states and molecules; higher values indicate that the post-training update (post-trained minus pretrained score, position component) points in a direction similar to the MLFF force. The distribution concentrates toward large cosine similarity, supporting the interpretation that alignment learns a force-like correction in the reverse dynamics.

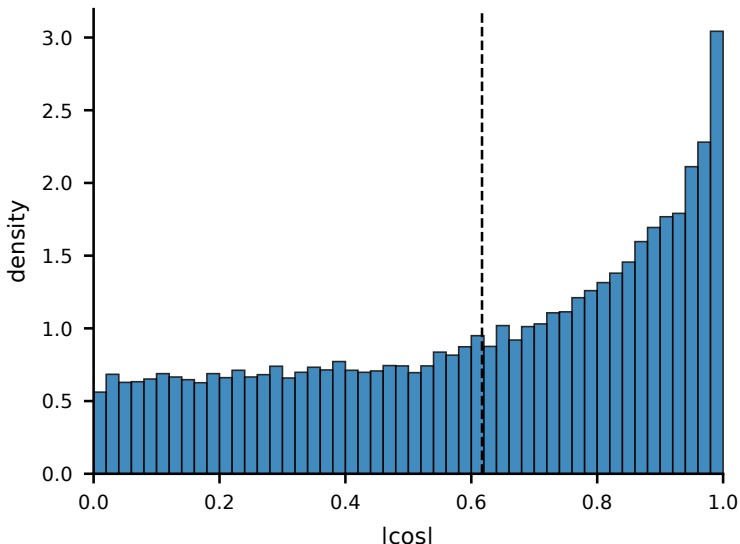

*Figure 11.* Distribution of $|\cos(\mathbf{F}_{\mathrm{alc}}, \mathbf{F}_{\phi})|$ between the position-space alchemical force $\mathbf{F}_{\mathrm{alc}}$ (post-trained minus pretrained score, Eq. 10) and the MLFF force $\mathbf{F}_{\phi}(\hat{z}_{0|t})$ (Eq. 11). Larger values indicate stronger directional agreement.

