# OpenReview forum: "Elign: Equivariant Diffusion Model Alignment from Foundational Machine Learned Force Fields"
_ICML.cc/2026/Conference — ICML 2026 regular_

### Official Review · Reviewer_SzcA · 2026-03-09

**Soundness:** 4
**Presentation:** 4
**Significance:** 3
**Originality:** 2
**Overall Recommendation:** 5
**Confidence:** 4

**Summary:**

This paper addresses the conformational distribution gap between low-fidelity training data used for molecular diffusion models and the distribution implied by higher-fidelity quantum mechanical methods. The authors use a transferable machine learning force fields (MLFF) trained on DFT-level data as a reward signal to correct the diffusion model via post-training reinforcement learning based on GRPO. This amortizes the cost of the MLFF at training time, eliminating the need for expensive energy evaluations during inference. The approach is evaluated on minimum-energy conformer generation using the QM9 and GEOM-Drugs benchmarks.

**Compliance With Llm Reviewing Policy:**

Affirmed.

**Final Justification:**

The additional evaluation in the rebuttal has considerably extended the scope and applicability of the method, alleviating my main concerns. This is a timely scientific work that addresses a significant scientific question in a convincing manner.

**Key Questions For Authors:**

- Did the authors try to apply their method to proper Boltzmann-distributed data?

**Limitations:**

yes

**Strengths And Weaknesses:**

Strengths:
- The motivation of this work, showing whether diffusion models pre-trained on low-fidelity data (e.g. empirical force field MD) could be post-trained to align with higher-fidelity MLFFs, is a timely and significant scientific question.
- The paper is well-written and easy to follow.
- The proposed multi-stage training approach is sound.
- The ablations are helpful and the DFT-based evaluation is solid.

Weaknesses:
- The authors motivate their method with Boltzmann-distributed data, but evaluate their method only on datasets containing energy-minimized conformations (it is a stretch to argue this is 0-temperature Boltzmann-distributed). Finite-temperature Boltzmann distributed data (even just dipeptides) would be far more interesting and would allow to answer the main hypothesis of the paper (see motivation above).
- Similarly, the metrics considered here for QM9 and GEOM-drug are fairly saturated, making it difficult to draw definitive empirical conclusions. These two benchmarks also have very limited practical significance.
- As the authors note themselves under related works, there are many existing works that proposed very similar approaches, including RL for diffusion post-training and diffusion guidance based on MLFFs.

---

> ### Author Rebuttal · Authors · 2026-03-28
>
> Thank you for your thoughtful feedback. We address your concerns below and describe the improvements we will make in our revision.
>
> **W1 + Q1.** We thank the reviewer for the insightful suggestion. We agree that the most important empirical gap in the original submission was the lack of a direct finite-temperature test. To respond to this concern, we added an alanine dipeptide experiment at 300K. We use a trajectory-level split: 36k training and 4k validation frames from one MD trajectory, and 20k test frames from an independent held-out run. We first pretrain a fixed-topology unconditional EDM generator over coordinates, and then post-train it with GRPO using a UMA energy reward. Results are reported using matched evaluation with 256 generated samples. Elign reduces Ramachandran $(\phi,\psi)$ JSD from 0.6432 to 0.5078 and energy $W_1$ from 0.831 to 0.254. We believe this is a critical rebuttal result, since it provides a more direct test of finite-temperature equilibrium.
>
> || Baseline | Elign | Baseline+Guidance |
> | :-----|:---- |:----|:-----|
> | $(\phi,\psi)$-JSD | 0.6432 | **0.5078**| 0.6231|
> | Energy $W_1$| 0.831| **0.254**| 0.782 |
>
> We also thank the reviewer for prompting a clearer theoretical interpretation. We note that Elign is not an exact de novo Boltzmann sampler. Instead, Theorem 1 shows that post-training Gibbs-tilts the pretrained terminal law. The alanine result should therefore be viewed not as exact Boltzmann recovery, but as evidence that post-training moves the learned distribution closer to Boltzmann.
>
> Lastly, we agree this also points to an interesting limitation and future direction. Extending the framework toward exact Boltzmann sampling would likely require accounting for the pretrained model’s implicit prior term (e.g. subtracting the prior energy contribution in reward). We are grateful to the reviewer for raising this issue, and we will make both the current interpretation and this future direction clearer in the revision.
>
> **W2.** We thank the reviewer for this thoughtful point. We agree that QM9 is a relatively mature benchmark, and we apologize for not making our positioning clearer. More broadly, we appreciate the reviewer encouraging us to clarify which evidence in the paper reflects benchmark improvement versus broader physical quality and generality.
>
> First, we would like to respectfully note that GEOM-Drugs remains meaningfully unsaturated. The pretrained EDM baseline achieves 81.3% atom stability and 91.9% validity on GEOM-Drugs, which contains substantially larger molecules, up to 181 atoms. After alignment, these improve to 87.94% and 99.40%, respectively. Thus, the gains on GEOM-Drugs are not marginal ceiling effects, but meaningful improvements on a larger and still challenging benchmark.
>
> Second, the original submission already evaluates physical quality beyond standard benchmark metrics. In particular, we include an independent DFT oracle evaluation, and Figure 4 shows favorable distributional shifts in both formation energy per atom and force RMS. This is important because it shows that the gains are not merely improvements on benchmark-style validity numbers, but correspond to improved physical quality under an oracle that is not used during training. We apologize for not emphasizing this more clearly.
>
> Third, because your comment also raises the broader question of generality, we added transfer experiments across backbones, oracles, and generative frameworks to test whether the effect is specific to one particular setup.
>
> ||Backbone|Oracle|Atom Stab. (baseline) % |Atom Stab. (Elign) %|
> |:---------- |:------|:---------|:----------|:------------|
> | Original| EDM| UMA-1p1-M  | 81.3| 87.94|
> | Backbone transfer|GeoLDM|UMA-1p1-M| 84.4| 88.92|
> | Oracle transfer| EDM|MACE-POLAR| 81.3| 86.98|
> | Flow transfer|EquiFM|UMA-1p1-M| 84.1| 87.25|
>
> The results show that the framework is transferable to various settings. We are grateful to the reviewer for raising this concern, and in the revision we will make the paper’s positioning clearer.
>
> **W3.** We agree that RL for diffusion and physical guidance both have prior art, so our contribution is not the novelty of these components alone. Instead, we frame Elign as a **portable post-training alignment layer** based on a **double-amortization** principle, and will revise the paper to make this clearer and add a comparison table to prior work.
>
> Concretely, the first amortization is at the reward level: a foundation MLFF replaces quantum calculations with a reusable surrogate. The second is at the generator level: post-training absorbs sampling-time guidance into the model, so inference remains oracle-free and as fast as the base sampler. Within this framework, Elign further improves the base generator through dense PBRS energy shaping and a disentangled GRPO objective for force and energy. It is also agnostic to backbones, reward models, and generator families. We thank the reviewer for prompting a clearer articulation of this point.

---

> > ### Author Rebuttal · Reviewer_SzcA · 2026-04-02
> >
> > The authors have addressed my main concerns well. I have raised my score accordingly.

---

### Official Review · Reviewer_CGkP · 2026-03-10

**Soundness:** 3
**Presentation:** 3
**Significance:** 3
**Originality:** 3
**Overall Recommendation:** 4
**Confidence:** 4

**Summary:**

This paper studies post-training alignment for equivariant diffusion models for molecular conformation generation. The main idea is to use a pretrained MLIP (UMA) as a preference model, which provides energy and atomic forces during reinforcement learning. This paper aims to make the reverse diffusion process generate structures with lower energy and smaller forces, expecting that the generated conformations will be more physically stable. The paper proposes a force- and energy-disentangled GRPO objective and uses potential-based reward shaping to improve training. Experiments on benchmark datasets show improved stability, competitive validity, and uniqueness.

**Compliance With Llm Reviewing Policy:**

Affirmed.

**Final Justification:**

I appreciate the authors’ thoughtful rebuttal and the additional experiments. Overall, the paper presents an interesting and practically meaningful idea: using a pretrained foundational MLFF as a preference model for post-training alignment of equivariant molecular generative models. The method is original, and the results show consistent improvements in stability-related metrics on QM9 and GEOM-Drugs.

The rebuttal addressed most of my main concerns. The added ablations and the extension to a flow-based model also strengthen the paper and make the main claim more convincing. My main remaining concern is about the breadth of validation. The additional experiments are helpful, but the newly included GeoLDM and EquiFM backbones are still relatively earlier models, so it remains unclear how well the proposed alignment strategy generalizes to more recent diffusion and flow-based molecular generative models.

Overall, my final recommendation remains unchanged.

**Key Questions For Authors:**

1) On the reverse diffusion trajectory. The reverse process is a learned denoising trajectory, not a real molecular dynamics trajectory, right? Can the reverse generation process be viewed as a multi-step relaxation process?

2) On the ablation study. This paper uses EDM as the generator and UMA as the preference model. However, it is unclear whether the gain mainly comes from post-training alignment or stronger MLIP integrated into the diffusion models. I suggest adding the following experiments.
Setting 1: Replace the EDM backbone with a stronger equivariant MLIP architecture, such as NequIP or MACE, and train the generative model from scratch. Setting 2: Keep EDM as the generator but use a pretrained NequIP or MACE model as the preference model for reinforcement learning alignment, similar to the current UMA-based setup. This comparison would make the paper much more convincing and would clarify where the performance gain really comes from.

3) On the generalization of this method. Whether this reinforcement learning idea also works for flow-based generative models. If the same idea can also improve a flow model, this paper would be more solid.

**Limitations:**

Yes

**Strengths And Weaknesses:**

Strengths： The main idea is interesting. Using a pretrained MLIP as a preference model is a practical way to inject physical knowledge into the generator. Although many diffusion models can generate reasonable 3D structures, physical stability remains a major concern.

Weaknesses： The paper assumes that the preference model is reliable in intermediate reverse states. However, these intermediate states can be highly noisy and unstable, and may not have been trained during the pretraining stage of foundation MLIPs. Only energy and force signals with high accuracy serve as trustworthy rewards during alignment.

---

> ### Author Rebuttal · Authors · 2026-03-28
>
> Thank you for your thorough review and constructive feedback. We appreciate your careful reading of our paper. We address your concerns below and outline specific improvements we will make in the revision. The following table summarizes the newer results on GEOM-Drugs atom stability:
>
> | Experiment                       | Before Elign | After Elign |
> | :------------------------------- | :----------: | :---------: |
> | GeoLDM + Elign, UMA-1p1-M reward |    84.4%     |   88.92%    |
> | EDM + Elign, MACE-POLAR reward   |    81.3%     |   86.98%    |
> | Flow + Elign, UMA-1p1-M reward   |    84.1%     |   87.25%    |
>
> **W1**. We thank the reviewer for raising this important point. We agree this is the key concern for the shaping component, and we apologize for not highlighting our design choices and safeguards more explicitly in the original submission. In Elign, the MLFF never sees raw noisy states; it only evaluates the **posterior-mean clean estimate** $\hat{z}\_{0|t}$. This mirrors common practice in vision diffusion fine-tuning: DRaFT (Clark et al., 2024) and ReFL (Xu et al., 2024) both compute reward on the predicted $\hat{x}\_{0|t}$ from noisy intermediates to update the denoiser. We adopt the same idea but apply it to MLFF energy rather than an image-quality model. We also skip the highest-noise prefix (700 steps on QM9, 600 on GEOM-Drugs), and we restrict force reward to the terminal step only because force-based shaping was unstable in practice.
>
>
> **Q1**. We thank the reviewer for this insightful clarification request. Correct. The reverse process is better understood as a **learned transport** whose terminal samples are steered toward lower energy and smaller residual force. The diffusion "time" index is a **noise level** that parameterizes this transport from prior to data; it has no correspondence to MD time, which evolves under Hamiltonian or Langevin dynamics with a physical timestep. We will state this distinction more explicitly in the revision.
>
> **Q2**. We are very grateful for this suggestion, which we found both concrete and highly helpful. We followed the spirit of both requested ablations (see the table above). _Setting 1_ (change the generator): replacing EDM with GeoLDM and applying Elign improves atom stability from 84.4% to 88.92%. _Setting 2_ (change the preference model): keeping the generator fixed and replacing UMA with MACE-POLAR still improves atom stability from 81.3% to 86.98%. These two experiments isolate the alignment effect from the original EDM + UMA pairing.
>
> **Q3**. We agree this is an important test of the generality of the method, and we appreciate the reviewer for pointing us toward a stronger validation. Following this suggestion, we extended Elign to a flow-based generator. Using EquiFM (Song et al., 2023) as the backbone, atom stability improves from **84.1%** to **87.25%** on GEOM-Drugs. This supports the view that the core post-training idea is not specific to diffusion models. In the revision, we will **update the manuscripts** to incorporate this conversion; we briefly outline the procedure below.  The adaptation involves converting the deterministic ODE into a **stochastic policy** over the tuned suffix. Following Liu et al. (2025), the general marginal-preserving SDE is:
>
>
>  $$\mathrm{d}x_t = \left(v_t(x_t) - \frac{\sigma_t^2}{2}\nabla\log p_t(x_t)\right)\mathrm{d}t + \sigma_t\,\mathrm{d}w.$$
>
>  For rectified flow, the score admits a closed-form expression in terms of the velocity field, giving
>
>  $$\mathrm{d}x_t = \left[v_t(x_t) + \frac{\sigma_t^2}{2t}\left(x_t + (1-t)\,v_t(x_t)\right)\right]\mathrm{d}t + \sigma_t\,\mathrm{d}w.$$
>
>  Euler–Maruyama discretization then yields the update rule
>
>  $$x_{t+\Delta t} = x_t + \left[v_\theta(x_t, t) + \frac{\sigma_t^2}{2t}\left(x_t + (1-t)\,v_\theta(x_t, t)\right)\right]\Delta t + \sigma_t\sqrt{\Delta t}\;\epsilon,$$
>
> which defines a Gaussian policy whose log-probabilities are available in closed form for GRPO. The shared prefix ($T \to T_{\mathrm{prefix}}$) is rolled out with the original deterministic ODE; only in the suffix ($T_{\mathrm{prefix}} \to 0$) do we use the stochastic form. At inference we set $\sigma_t = 0$ and recover the deterministic flow sampler end-to-end. We appreciate the reviewer’s suggestion here in particular, because it led to a  stronger empirical case for the method, and we will highlight this extension more clearly in the revision.
>
> [1]: Clark K et al. Directly Fine-Tuning Diffusion Models on Differentiable Rewards. ICLR 2024.
>
> [2]: Xu J et al. Imagereward: Learning and evaluating human preferences for text-to-image generation. NeurIPS 2023.
>
>
> [3] Liu J et al, Training Flow Matching Models via Online RL. NeurIPS 2025.
>
> [4]  Song Y et al. Equivariant Flow Matching with Hybrid Probability Transport for 3D Molecule Generation. NeurIPS 2023.

---

> > ### Author Rebuttal · Reviewer_CGkP · 2026-04-02
> >
> > The authors have addressed most of my concerns. However, both GeoLDM and EquiFM are relatively earlier works from 2023. It would be helpful to see whether the proposed method can also be validated on more recent generative and flow-based models. At this point, I am inclined to keep my score unchanged.

---

> > > ### Author Response · Authors · 2026-04-06
> > >
> > > Thank you for this thoughtful and constructive follow-up. We fully agree that testing on newer backbones is an especially insightful way to strengthen the generality claim, and we are grateful for the reviewer's guidance here. In response, we extended Elign beyond GeoLDM and EquiFM to the more recent UniGem model. Elign improves GEOM-Drugs atom stability on UniGem [1] from **84.4%** to **88.51%** after alignment, showing the gains are not specific to the original EDM + UMA combination and could potentially continue to hold on a newer pretrained model. Together with our MACE-POLAR reward-model ablation and flow-based results on EquiFM, the revised experiments now support the alignment effect along three complementary directions: reward model, generator backbone, and generator family. We sincerely appreciate the reviewer's suggestion, which meaningfully strengthened the empirical case, and we will highlight this expanded evidence more clearly in the revision.
> > >
> > > [1]: Feng S, Ni Y, Lu Y, et al. Unigem: A unified approach to generation and property prediction for molecules. ICLR 2025.

---

### Official Review · Reviewer_jndb · 2026-03-11

**Soundness:** 3
**Presentation:** 3
**Significance:** 2
**Originality:** 2
**Overall Recommendation:** 4
**Confidence:** 3

**Summary:**

Elign is a multi-stage computational framework designed to align three-dimensional molecular diffusion models with fundamental physical principles through post-training reinforcement learning. The approach implements a double amortization strategy to resolve the conflict between structural fidelity and generation speed. Initially, the method amortizes expensive quantum chemical calculations by utilizing a foundational machine learning force field to act as a learned preference model for providing physical signals. Subsequently, it eliminates the need for repeated runtime queries by shifting physical steering into a dedicated training phase where the reverse diffusion policy is fine-tuned. This optimization is conducted via Force-Energy Disentangled Group Relative Policy Optimization, which formulates the denoising process as a finite-horizon Markov decision process. By integrating potential-based energy rewards with force-based stability rewards, the sampler learns to prioritize configurations that exhibit both thermodynamic stability and mechanical equilibrium.

**Compliance With Llm Reviewing Policy:**

Affirmed.

**Final Justification:**

I maintain a **Weak Accept**. The method appears technically sound and practically useful: amortizing expensive quantum evaluations with a pretrained ML force field and shifting physical steering from inference-time guidance to RL fine-tuning can improve energy/force/stability while keeping sampling speed unchanged.

The rebuttal addressed my main technical concerns and increased my confidence in the implementation (no backprop through long rollouts; group-relative normalization done within shared-prefix groups rather than across unrelated molecules; clarification on discrete atom types and variance reduction). It also clarified the intended support-preserving nature of the alignment and practical safeguards against MLFF bias amplification and shaping instability. Overall, the rebuttal reinforced my prior assessment on soundness and clarity of specific points, but did not change my evaluation of originality and significance.

My remaining reservations are that the overall idea feels not very novel, and the current evaluation does not convincingly demonstrate broad generalization, so I expect the impact on the community to be limited despite the solid engineering and results.

**Key Questions For Authors:**

1. I was wondering how memory is managed during the parallel rollouts, particularly when storing intermediate activations for backpropagation through long denoising trajectories.
2. Could you explain how group-relative statistics are standardized when molecules of different sizes and energy scales are included in the same batch?
3. Since the molecular state includes discrete atom types, I would be interested to know if any specific techniques were used to reduce variance in the policy gradient during training.

**Limitations:**

yes

**Strengths And Weaknesses:**

Strengths:
1. The model maintains the same inference speed as unguided samplers because no physics oracles are queried during generation.
2. Potential-based reward shaping converts sparse terminal rewards into dense signals that facilitate optimization over long horizons.
3. The dual reward system of energy and force signals improves both thermodynamic stability and mechanical equilibrium.

Weaknesses:
1. The alignment objective requires the terminal law to be absolutely continuous with respect to the pretrained law, which restricts optimization to the manifold learned during pre-training. This constraint may prevent the model from discovering stable configurations in chemical regions that the base model never observed.
2. Theorem 2 establishes that distributional error is sensitive to the product of force field bias and effective inverse temperature. Consequently, aggressive alignment toward low-energy states can inadvertently amplify small biases present in the machine learning force field proxy.
3. Reward shaping relies on plug-in estimates of clean geometry derived from noisy intermediate states using the diffusion posterior mean. These estimates are inherently unstable and prone to high variance during high-noise diffusion steps.

---

> ### Author Rebuttal · Authors · 2026-03-28
>
> Thank you for your thorough review and constructive feedback. We appreciate your careful reading of our paper. We address your concerns below and outline specific improvements we will make in the revision.
>
> **W1**. We thank the reviewer for this insightful comment, which helps clarify an important aspect of the method. We agree that Elign is a **support-preserving** alignment method rather than an unconstrained discovery method, and this is an intentional design choice. Indeed, Theorem 1 shows exactly this: the optimal terminal law is a Gibbs reweighting of the pretrained terminal law. We view this as a feature rather than defect because it improves physical fidelity while preserving the structural prior and validity inherited from the base generator. In this respect, Elign parallels current practice in large language models, where
> post-training is likewise constrained by the support provided by next-token-prediction
> pretraining.  Moreover, the new alanine dipeptide result shows that this constrained reweighting is still scientifically meaningful in the finite-temperature regime, reducing the held-out $(\phi,\psi)$-JSD from 0.6432 to 0.5078. In the revision, we will highlight this point more explicitly and better explain that support preservation is a deliberate feature of the framework.
>
>  **W2.** We thank the reviewer for this observation. It is a valid and important reading of the theorem that we should have emphasized more clearly. In practice, we guard against this failure mode by maintaining a nonzero KL penalty against the pretrained policy, which directly controls the effective inverse temperature and prevents the terminal law from tilting aggressively enough to amplify MLFF biases. Additionally, the theorem highlights the importance of improving the base pretrained model itself, and higher-quality pretraining directly reduces the risk of bias amplification. We will make this point explicit in the revision.
>
> **W3**.  We agree this is the principal risk for the shaping component. Three design choices address it in our implementation. **First**, the MLFF is never queried on raw noisy $z_t$; it only sees the posterior-mean clean estimate $\hat{z}\_{0|t}$.  Computing a reward on this clean estimate is standard practice in vision diffusion alignment: DRaFT (Clark et al., 2024) and ReFL (Xu et al., 2024) both apply alignment signal to predicted $\hat{z}\_{0|t}$ to refine the denoiser mid-trajectory. We adopt the same idea but apply it to MLFF energy rather than an image-quality model. **Second**, shaping is disabled for the highest-noise prefix: we skip 700 steps on QM9 and 600 on GEOM-Drugs, so dense shaping is applied only in the relatively denoised regime. _Third_, we found force-based shaping to be unstable and therefore restrict force reward to the terminal step only. We will make these choices and their rationale more explicit in the revision.
>
> **Q1**. We appreciate the reviewer raising this question, which helps clarify the method.  In the framework, there is **no backpropagation** through the rollout generation process. Rollouts are sampled under the frozen old policy, reward computation is stop-gradient, and PPO recomputes only per-step log-probabilities on the sampled state–action pairs. In practice, MLFF evaluation is chunked, rollout generation is micro-batched (24 on QM9, 32 on GEOM-Drugs), and the policy update uses micro-batches with gradient accumulation. The dominant memory cost is therefore storing the rollout trajectories — states and per-step log-probabilities — rather than backpropagating through the SDE solver. We apologize for the unclarity and we will add a paragraph right after Sec. 3.3 to highlight this.
>
> **Q2**.  We appreciate the reviewer raising this question. In our implementation, GRPO is never performed across unrelated molecules. Each group consists of $K$ continuations branched from the same cached prefix state $z_{T_{\mathrm{prefix}}}$, and all rollouts share the same atom count and conditioning context. Within each group, the force channel uses per-atom RMS aggregation, while the energy channel, drawing parallel to an MLFF, is standardized after subtracting atom reference energies. In the revision, we will add a specific paragraph in Sec. 3.1 to highlight this.
>
> **Q3**. We appreciate this question, as it improves the technical clarity of the paper. Discrete atom types are represented as relaxed continuous feature channels in EDM, so the reverse kernel is continuous/Gaussian throughout. The main variance-reduction mechanisms are shared-prefix grouping, disentangled normalization, advantage clipping, and KL regularization. This is also supported by the ablation in Table 10: removing shared-prefix grouping substantially degrades molecule stability and $V\times U$.
> [1]: Clark K et al. Directly Fine-Tuning Diffusion Models on Differentiable Rewards. ICLR 2024.
>
> [2]: Xu J et al. Imagereward: Learning and Evaluating Human Preferences for Text-to-Image Generation. NeurIPS 2023.

---

> > ### Author Rebuttal · Reviewer_jndb · 2026-04-02
> >
> > Thank you, my concerns have been addressed.

---

### Official Review · Reviewer_pDWa · 2026-03-13

**Soundness:** 3
**Presentation:** 1
**Significance:** 2
**Originality:** 3
**Overall Recommendation:** 4
**Confidence:** 3

**Summary:**

The paper proposes a method to improve the generation of 3D molecular conformations/ molecule. The method utilizes the MLFF to give guidance for the reverse diffusion during sampling, which is also modeled as a reinforcement learning problem. The experimental results on small-molecule generation showcase the effectiveness of the proposed method.

**Compliance With Llm Reviewing Policy:**

Affirmed.

**Final Justification:**

The authors addressed my concerns. I improved my score.

**Key Questions For Authors:**

I wonder whether the method can be used for conformation generation or molecular dynamics generation, such as MDgen or Bioemu.

**Limitations:**

yes

**Strengths And Weaknesses:**

Strengths:
Using the LLM-similar pipeline to improve the molecular generative model's generative ability in terms of structural stability sounds interesting and promising.

Weaknesses:
- The paper is hard to read. 1. Too many abbreviations; 2. Too many mathematical symbols 3. The description of the technical details is too complex， like sec 3.2. The authors can only put the consequence in the main paper to improve readability.
- The experimental improvement is marginal. QM9 and the drugs dataset are small (#atoms). Many baselines achieve very good performance on the metrics. So the limited experimental results on these two datasets are not very convincing. The authors can consider using more datasets to better support the methods, such as protein.
- The proposed pipeline will introduce many additional operations and will also use the big model UMA. So the reader will care about the sampling time if they use the proposed method (compared to w.o guidance). In addition, the model relies on the pretrained UMA, so it would also be important to know the proposed method's sensitivity to the UMA.  Actually, the author can consider using the ground truth force and energy instead of predicting by UMA to better verify the robustness of the proposed method. Also, EDM can be replaced with another optional backbone to make the proposed pipeline convincing.

---

> ### Author Rebuttal · Authors · 2026-03-28
>
> Thank you for your detailed review and helpful feedback. We address your concerns below and describe the improvements we will make in our revision.  We also added four new experiments, and the results are summarized in the table below:
> | Setting | Metric|Before Elign|After Elign|
> | :---------------| :---------------| :------------: | :----------: |
> | GeoLDM + Elign, UMA-1p1-M reward | GEOM-Drugs atom stability|84.4%|**88.92%**|
> | EDM + Elign, MACE-POLAR reward   | GEOM-Drugs atom stability|81.3%| **86.98%**|
> | Flow + Elign, UMA-1p1-M reward| GEOM-Drugs atom stability|84.1%|**87.25%**|
> | Alanine dipeptide| $(\phi,\psi)$-JSD / energy $W_1$ | 0.6432 / 0.831 | **0.5078 / 0.254** |
>
> **W1**. We agree that the paper can be easier to read. In the revision, we will reduce abbreviations, move most of the symbol-heavy material from Sec. 2-3 to the appendix, and start with a more intuitive pipeline-level explanation before introducing the full notation.
>
> **W2**. We thank the reviewer for this thoughtful and important point, which has meaningfully strengthened the paper.  We agree that QM9 alone would not be sufficient, and we do not want the paper to rest on that benchmark alone. During the rebuttal, we therefore added broader tests: **generator-backbone transfer, reward-model transfer, sampler-family transfer**, and a **finite-temperature equilibrium evaluation** on alanine dipeptide.
>
> At the same time, we would like to respectfully note that the original benchmarks already show notable improvements on physically meaningful metrics **relative to the pretrained baseline**. On QM9, Elign improves molecule stability from 82.00% to 93.70%. On GEOM-Drugs, Elign improves atom stability from 81.3% to 87.94%.  GEOM-Drugs is also more challenging than QM9, with larger and more flexible molecules (up to 181 atoms, 44.2 on average), and the pretrained baselines remain well below ceiling on atom stability.
>
> Most importantly, the new rebuttal results go beyond standard minimum-energy conformer benchmarks. On alanine dipeptide, evaluated against an independent held-out equilibrium MD trajectory, Elign reduces Ramachandran $(\phi,\psi)$-JSD from 0.6432 to 0.5078 and energy $W_1$ from 0.831 to 0.254. We hope this directly addresses the concern that the original evaluation was too narrow. We view this as the right first biomolecular test because it targets equilibrium-distribution fidelity rather than only minimized structures. Full protein-scale experiments are an important next step, but we do not want to overclaim beyond what we could rigorously test during the rebuttal period.
>
> **W3**.
> **Sampling time**.  We apologize for not conveying this more clearly. At sampling time, Elign makes zero runtime oracle calls and uses the same sampler as the base model, so the cost is unchanged.  Elign achieves an 8-16x speedup over runtime guidance on 60–200-atom systems.
>
> **Sensitivity to the reward model.** Within UMA, changing the reward model from UMA-1p1-M to UMA-1p1-S changes the stability/diversity tradeoff on QM9 in an interpretable way: molecule stability improves from 93.70% to 94.83%, while $V \times U$ decreases from 95.31% to 90.81% (Main-text Table 2)). To verify that the method is not tied to UMA at all, we ran reward-model transfer with EDM fixed and MACE-POLAR used as the reward model. GEOM-Drugs atom stability still improves from 81.3% to 86.98%.
>
> **Verification against ground-truth physics.** We note that the original paper already includes a baseline that uses DFT as the reward: RLPF (Zhou et al., 2025) trains with DFT-level energy evaluations, which is the closest practical equivalent to using ground-truth force and energy. We apologize for not highlighting this more clearly. Elign matches or improves the main stability metrics on QM9 relative to RLPF while avoiding DFT queries during training. Additionally, the original paper evaluates the aligned model with an independent DFT oracle and finds lower DFT force RMS and lower energy, which indicates that the gains are not simply exploiting the MLFF reward.
>
> **Generator transfer.** The method is not tied to EDM. With GeoLDM as the backbone, atom stability improves from 84.4% to 88.92%, and with a flow-matching model it improves from 84.1% to 87.25%.
>
> We thank the reviewer for this valuable suggestion, which has improved the clarity of the method.  We will revise Section 4 to highlight cost and RLPF comparisons, and add reward-model and generator transfer tables.
>
> **Q1**. We thank the reviewer for this helpful suggestion. We agree extending Elign to models such as MDGen or BioEmu is a concrete future direction, which we will note in the discussion. The core post training loop is largely architecture agnostic: it rolls out the reverse process, scores final samples with a reward model, and updates the generator with GRPO. In principle, this applies to any generative model whose reverse dynamics can be viewed as a sequential policy, so we do not see an architectural barrier there.

---

> > ### Author Rebuttal · Reviewer_pDWa · 2026-03-31
> >
> > The authors addressed my concerns. I improved my score. The writing and presentation are not satisfactory, so I just Weak Accept.

---

### Decision · Program_Chairs · 2026-04-30

**Decision:**

Accept (regular)

**Comment:**

In this submission, the authors post-trained an equivariant diffusion model with machine learning-based force fields, which leads to better generation results of ground-state molecular conformations. All reviewers acknowledged the contributions of this work and three of them leaned towards weak acceptance. AC agreed with this decision.